# VARIATIONAL IMBALANCED REGRESSION

## ABSTRACT

Existing regression models tend to fall short in both accuracy and uncertainty estimation when the label distribution is imbalanced. In this paper, we propose a probabilistic deep learning model, dubbed variational imbalanced regression (VIR), which not only performs well in imbalanced regression but naturally produces reasonable uncertainty estimation as a byproduct. Different from typical variational autoencoders assuming I.I.D. representations (a data point's representation is not directly affected by other data points), our VIR borrows data with similar regression labels to compute the latent representation's variational distribution; furthermore, different from deterministic regression models producing point estimates, VIR predicts the entire normal-inverse-gamma distributions and modulates the associated conjugate distributions to impose probabilistic reweighting on the imbalanced data, thereby providing better uncertainty estimation. Experiments in several real-world datasets show that our VIR can outperform state-of-the-art imbalanced regression models in terms of both accuracy and uncertainty estimation.

## 1 INTRODUCTION

Deep regression models are currently the state of the art in making predictions in a continuous label space and have a wide range of successful applications in computer vision (Yin et al., 2021), natural language processing (Jiang et al., 2020), etc. However, these models fail however when the label distribution in training data is imbalanced. For example, in visual age estimation (Moschoglou et al., 2017), where a model infers the age of a person given her visual appearance, models are typically trained on imbalanced datasets with overwhelmingly more images of younger adults, leading to poor regression accuracy for images of children or elderly people (Yang et al., 2021). Such unreliability in imbalanced regression settings motivates the need for both *improving performance for the minority* in the presence of imbalanced data and, more importantly, *providing reasonable uncertainty estimation* to inform practitioners on how reliable the predictions are (especially for the minority where accuracy is lower).

Existing methods for deep imbalanced regression (DIR) only focus on improving the accuracy of deep regression models by smoothing the label distribution and reweighting data with different labels (Yang et al., 2021). On the other hand, methods that provide uncertainty estimation for deep regression models operates under the balance-data assumption and therefore do not work well in the imbalanced setting (Amini et al., 2020; Mi et al., 2022; Charpentier et al., 2022).

To simultaneously cover these two desiderata, we propose a probabilistic deep imbalanced regression model, dubbed variational imbalanced regression (VIR). Different from typical variational autoencoders assuming I.I.D. representations (a data point's representation is not directly affected by other data points), our VIR assumes Neighboring and Identically Distributed (N.I.D.) and borrows data with similar regression labels to compute the latent representation's variational distribution. Specifically, VIR first encodes a data point into a probabilistic representation and then mix it with neighboring representations (i.e., representations from data with similar regression labels) to produce its final probabilistic representation; VIR is therefore particularly useful for minority data as it can borrow probabilistic representations from data with similar labels (and naturally weigh them using our probabilistic model) to counteract data sparsity. Furthermore, different from deterministic regression models producing point estimates, VIR predicts the entire normal-inverse-gamma distributions and modulates the associated conjugate distributions by the importance weight computed from the smoothed label distribution to impose probabilistic reweighting on the imbalanced data. This allows the negative log likelihood to naturally put more focus on the minority data, thereby balancing the

accuracy for data with different regression labels. Our VIR framework is compatible with any deep regression models and can be trained end to end.

We summarize our contributions as below:

1. While previous work has studied imbalanced regression and uncertainty estimation *separately*, none of them has considered uncertainty estimation in the imbalanced setting. We identify the problem of probabilistic deep imbalanced regression as well as two desiderata, balanced accuracy and uncertainty estimation, for the problem.

2. We propose VIR to simultaneously cover these two desiderata and achieve state-of-the-art performance compared to existing methods.

3. As a byproduct, we also provide strong baselines for benchmarking high-quality uncertainty estimation and promising prediction performance on imbalanced datasets.

## 2  RELATED WORK

**Variational Autoencoder.** Variational autoencoder (VAE) (Kingma & Welling, 2014) is an unsupervised learning model that aims to infer probabilistic representations from data. However, as shown in Figure 1, VAE typically assumes I.I.D. representations, where a data point's representation is not directly affected by other data points. In contrast, our VIR borrows data with similar regression labels to compute the latent representation's variational distribution.

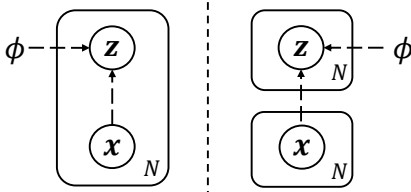

Figure 1: Comparison on inference networks between typical VAE (Kingma & Welling, 2014) and our VIR. In VAE (**left**), a data point's latent representation (i.e. $\mathbf{z}$) is affected only by itself, while in VIR (**right**), neighbors participate to modulate the final representation.

**Imbalanced Regression.** Imbalanced regression is underexplored in the machine learning community. Most existing methods for imbalanced regression are direct extensions of the SMOTE algorithm (Chawla et al., 2002), a commonly used algorithm for imbalanced classification, where data from the minority classes is over-sampled. These algorithms usually synthesize augmented data for the minority regression labels by either interpolating both inputs and labels (Torgo et al., 2013) or adding Gaussian noise (Branco et al., 2017; 2018).

Such algorithms fail to the distance in continuous label space and fall short in handling high-dimensional data (e.g., images and text). Recently, DIR (Yang et al., 2021) addresses these issues by applying kernel density estimation to smooth and reweight data on the continuous label distribution, achieving state-of-the-art performance. However, DIR only focuses on improving the accuracy, especially for the data with minority labels, and therefore does not provide uncertainty estimation, which is crucial to assess the predictions' reliability. Ren et al. (2022) focuses on re-balancing the mean squared error (MSE) loss for imbalanced regression, and Gong et al. (2022) introduces ranking similarity for improving deep imbalanced regression. In contrast, our VIR provides a principled probabilistic approach to simultaneously achieve these two desiderata, not only improving upon DIR in terms of performance but also producing reasonable uncertainty estimation as a much-needed byproduct to assess model reliability. There is also related work on imbalanced classification (Deng et al., 2021), which is related to our work but focusing on classification rather than regression.

**Uncertainty Estimation in Regression.** There has been renewed interest in uncertainty estimation in the context of deep regression models (Kendall & Gal, 2017; Kuleshov et al., 2018; Song et al., 2019; Zelikman et al., 2020; Amini et al., 2020; Mi et al., 2022; van Amersfoort et al., 2021; Liu et al., 2020; Gal & Ghahramani, 2016; Stadler et al., 2021; Snoek et al., 2019; Heiss et al., 2022). Most existing methods either directly predict the variance of the output distribution as the estimated uncertainty (Kendall & Gal, 2017; Zhang et al., 2019; Amini et al., 2020) or rely on post-hoc confidence interval calibration (Kuleshov et al., 2018; Song et al., 2019; Zelikman et al., 2020). Meanwhile, Posterior Networks methods Charpentier et al. (2020; 2022); Stadler et al. (2021) consider conjugate distribution, pseudo-count interpretations, posterior updates, and variational losses for fast and high-quality uncertainty estimation. Closest to our work is Deep Evidential Regression (DER) (Amini et al., 2020), which attempts to estimate both aleatoric and epistemic uncertainty (Kendall & Gal, 2017; Hüllermeier & Waegeman, 2019) on regression tasks by training

the neural networks to directly infer the parameters of the evidential distribution, thereby producing uncertainty measures. While Posterior Networks Charpentier et al. (2020; 2022) are designed for general classification/regression tasks and achieve promising performance, they do not explicitly consider imbalance in regression tasks, which is the focus of this paper. DER (Amini et al., 2020) is designed for the data-rich regime and therefore fails to reasonably estimate the uncertainty if the data is imbalanced; for data with minority labels, DER (Amini et al., 2020) tends produce unstable distribution parameters, leading to poor uncertainty estimation (as shown in Sec. 4). In contrast, our proposed VIR explicitly handles data imbalance in the continuous label space to avoid such instability; VIR does so by modulating both the representations and the output conjugate distribution parameters according to the imbalanced label distribution, allowing training/inference to proceed as if the data is balance and leading to better performance as well as uncertainty estimation (as shown in Sec. 4).

## 3 METHOD

In this section we introduce the problem setting, provide an overview of our VIR, and then describe details on each of VIR's key components.

### 3.1 PROBLEM SETTINGS

Assuming an imbalanced dataset in continuous space $\{\mathbf{x}_i, y_i\}_{i=1}^N$ where $N$ is the total number of data points, $\mathbf{x}_i \in \mathbb{R}^d$ is the input, and $y_i \in \mathcal{Y} \subset \mathbb{R}$ is the corresponding label from a continuous label space $\mathcal{Y}$. In practice, $\mathcal{Y}$ is partitioned into B equal-interval bins $[y^{(0)}, y^{(1)}), [y^{(2)}, y^{(2)}), ..., [y^{(B-1)}, y^{(B)})$, with slight notation overload. To directly compare with baselines, we use the same grouping index for target value $b \in \mathcal{B}$ as in (Yang et al., 2021).

We denote representations as $\mathbf{z}_i$, and use $(\widetilde{\mathbf{z}}_i^\mu, \widetilde{\mathbf{z}}_i^\Sigma) = q_\phi(\mathbf{z}|\mathbf{x}_i; \theta)$ to denote the probabilistic representations for input $\mathbf{x}_i$ generated by a probabilistic encoder parameterized by $\theta$. Similarly we use $(\widehat{y}_i, \widehat{s}_i)$ to denote the mean and variance of the predictive distribution generated by a probabilistic predictor $p_\theta(y_i|\mathbf{z})$. Furthermore, we denote $\bar{\mathbf{z}}$ as the mean of representation $\mathbf{z}_i$ in each bins (i.e., letting $\bar{\mathbf{z}} = \frac{1}{N_b} \sum_{i=1}^{N_b} \mathbf{z}_i$ in a bin with $N_b$ data points).

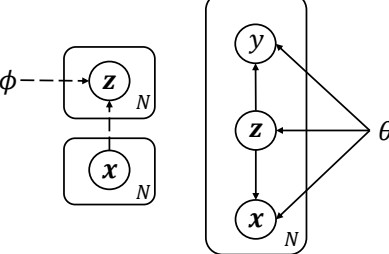

### 3.2 METHOD OVERVIEW

In order to achieve both desiderata in probabilistic deep imbalanced regression (i.e., performance improvement and uncertainty estimation), our proposed variational imbalanced regression (VIR) operates on both the encoder $q_\phi(\mathbf{z}_i|\{\mathbf{x}_i\}_{i=1}^N)$ and the predictor $p_\theta(y_i|\mathbf{z}_i)$.

Figure 2: Overview of our VIR method. **Left:** The inference model infers the latent representations given input $\mathbf{x}$'s in the neighborhood. **Right:** The generative model reconstructs the input and predicts the label distribution (including the associated uncertainty) given the latent representation.

Typical VAE (Kingma & Welling, 2014) lower-bounds input $\mathbf{x}_i$'s marginal likelihood; in contrast, VIR lower-bounds the marginal likelihood of input $\mathbf{x}_i$ and labels $y_i$:

$$\log p_\theta(\mathbf{x}_i, y_i) = \mathcal{D}_{\mathcal{KL}}\big(q_\phi(\mathbf{z}_i|\{\mathbf{x}_i\}_{i=1}^N)||p_\theta(\mathbf{z}_i|\mathbf{x}_i, y_i)\big) + \mathcal{L}(\theta, \phi; \mathbf{x}_i, y_i).$$

Note that our variational distribution $q_\phi(\mathbf{z}_i|\{\mathbf{x}_i\}_{i=1}^N)$ (1) does not conditions on labels $y_i$, since the task is to predict $y_i$ and (2) conditions on all (neighboring) inputs $\{\mathbf{x}_i\}_{i=1}^N$ rather than just $\mathbf{x}_i$. The second term $\mathcal{L}(\theta, \phi; \mathbf{x}_i, y_i)$ is VIR's evidence lower bound (ELBO), which is defined as:

$$\mathcal{L}(\theta, \phi; \mathbf{x}_i, y_i) = \underbrace{\mathbb{E}_q\big[\log p_\theta(\mathbf{x}_i|\mathbf{z}_i)\big]}_{\mathcal{L}_i^\mathcal{P}} + \underbrace{\mathbb{E}_q\big[\log p_\theta(y_i|\mathbf{z}_i)\big]}_{\mathcal{L}_i^\mathcal{P}} - \underbrace{\mathcal{D}_{\mathcal{KL}}(q_\phi(\mathbf{z}_i|\{\mathbf{x}_i\}_{i=1}^N)||p_\theta(\mathbf{z}_i))}_{\mathcal{L}_i^{\mathcal{KL}}}. \quad (1)$$

where the $p_\theta(\mathbf{z}_i)$ is the standard Gaussian prior $\mathcal{N}(\mathbf{0}, \mathbf{I})$, following typical VAE (Kingma & Welling, 2014), and the expectation is taken over $q_\phi(\mathbf{z}_i|\{\mathbf{x}_i\}_{i=1}^N)$, which infers $\mathbf{z}_i$ by borrowing data with similar regression labels to produce the balanced probabilistic representations, which is beneficial especially for the minority (see Sec. 3.3 for details).

Different from typical regression models which produce only point estimates for $y_i$, our VIR's predictor, $p_\theta(y_i|\mathbf{z}_i)$, directly produces the parameters of the entire NIG distribution for $y_i$ and further imposes probabilistic reweighting on the imbalanced data, thereby producing balanced predictive distributions (more details in Sec. 3.4).

## 3.3 Constructing $q(\mathbf{z}_i|\{\mathbf{x}_i\}_{i=1}^N)$

To cover both desiderata, one needs to (1) produce *balanced* representations to improve performance for the data with minority labels and (2) produce *probabilistic* representations to naturally obtain reasonable uncertainty estimation for each model prediction. To learn such *balanced probabilistic representations*, we construct the encoder of our VIR (i.e., $q_\phi(\mathbf{z}_i|\{\mathbf{x}_i\}_{i=1}^N)$) by (1) first encoding a data point into a **probabilistic representation**, (2) computing **probabilistic statistics** from neighboring representations (i.e., representations from data with similar regression labels), and (3) producing the final representations via **probabilistic whitening and recoloring** using the obtained statistics.

**Probabilistic Representations.** We first encode each data point into a probabilistic representation. Note that this is in contrast to existing work (Yang et al., 2021) that uses deterministic representations. We assume that each encoding $\mathbf{z}_i$ is a Gaussian distribution with parameters $\{\mathbf{z}_i^\mu, \mathbf{z}_i^\Sigma\}$, which are generated from the last layer in the deep neural network.

**From I.I.D. to Neighboring and Identically Distributed (N.I.D.).** Typical VAE (Kingma & Welling, 2014) is an unsupervised learning model that aims to learn a variational representation from latent space to reconstruct the original inputs under the I.I.D. assumption; that is, in VAE, the latent value (i.e., $\mathbf{z}_i$) is generated from its own input $\mathbf{x}_i$. This I.I.D. assumption works well for data with majority labels, but significantly harms performance for data with minority labels. To address this problem, we replace the I.I.D. assumption with the N.I.D. assumption; specifically, VIR's variational latent representations still follow Gaussian distributions (i.e., $\mathcal{N}(\mathbf{z}_i^\mu, \mathbf{z}_i^\Sigma)$, but these distributions will be first calibrated using data with neighboring labels. For a data point $(\mathbf{x}_i, y_i)$ where $y_i$ is in the $b$'th bin, i.e., $y_i \in [y^{(b-1)}, y^{(b)})$, we compute $q(\mathbf{z}_i|\{\mathbf{x}_i\}_{i=1}^N) \triangleq \mathcal{N}(\mathbf{z}_i; \widetilde{\mathbf{z}}_i^\mu, \widetilde{\mathbf{z}}_i^\Sigma)$ as

$$\text{Mean and Covariance of Initial } \mathbf{z}_i: \mathbf{z}_i^\mu, \mathbf{z}_i^\Sigma = \mathcal{I}(\mathbf{x}_i), \tag{2}$$

$$\text{Statistics of Bin } b\text{'s Statistics: } \boldsymbol{\mu}_b^\mu, \boldsymbol{\mu}_b^\Sigma, \boldsymbol{\Sigma}_b^\mu, \boldsymbol{\Sigma}_b^\Sigma = \mathcal{A}(\{\mathbf{z}_i^\mu, \mathbf{z}_i^\Sigma\}_{i=1}^N), \tag{3}$$

$$\text{Smoothed Statistics of Bin } b\text{'s Statistics: } \widetilde{\boldsymbol{\mu}}_b^\mu, \widetilde{\boldsymbol{\mu}}_b^\Sigma, \widetilde{\boldsymbol{\Sigma}}_b^\mu, \widetilde{\boldsymbol{\Sigma}}_b^\Sigma = \mathcal{S}(\{\boldsymbol{\mu}_b^\mu, \boldsymbol{\mu}_b^\Sigma, \boldsymbol{\Sigma}_b^\mu, \boldsymbol{\Sigma}_b^\Sigma\}_{b=1}^B), \tag{4}$$

$$\text{Mean and Covariance of Final } \mathbf{z}_i: \widetilde{\mathbf{z}}_i^\mu, \widetilde{\mathbf{z}}_i^\Sigma = \mathcal{F}(\mathbf{z}_i^\mu, \mathbf{z}_i^\Sigma, \boldsymbol{\mu}_b^\mu, \boldsymbol{\mu}_b^\Sigma, \boldsymbol{\Sigma}_b^\mu, \boldsymbol{\Sigma}_b^\Sigma, \widetilde{\boldsymbol{\mu}}_b^\mu, \widetilde{\boldsymbol{\mu}}_b^\Sigma, \widetilde{\boldsymbol{\Sigma}}_b^\mu, \widetilde{\boldsymbol{\Sigma}}_b^\Sigma),$$

where the details of functions $\mathcal{I}(\cdot)$, $\mathcal{A}(\cdot)$, $\mathcal{S}(\cdot)$, and $\mathcal{F}(\cdot)$ are described below.

**Function $\mathcal{I}(\cdot)$: From Deterministic to Probabilistic Statistics.** Different from deterministic statistics in (Yang et al., 2021), our VIR's encoder uses *probabilistic statistics* (i.e., *statistics of statistics*). Specifically, VIR treats $\mathbf{z}_i$ as a distribution with the mean and covariance $(\mathbf{z}_i^\mu, \mathbf{z}_i^\Sigma) = \mathcal{I}(\mathbf{x}_i)$ rather than a deterministic vector. As a result, all the deterministic statistics, $\boldsymbol{\mu}_b$, $\boldsymbol{\Sigma}_b$, $\widetilde{\boldsymbol{\mu}}_b$, and $\widetilde{\boldsymbol{\Sigma}}_b$ are replaced by distributions with the means and covariances, $(\boldsymbol{\mu}_b^\mu, \boldsymbol{\mu}_b^\Sigma)$, $(\boldsymbol{\Sigma}_b^\mu, \boldsymbol{\Sigma}_b^\Sigma)$, $(\widetilde{\boldsymbol{\mu}}_b^\mu, \widetilde{\boldsymbol{\mu}}_b^\Sigma)$, and $(\widetilde{\boldsymbol{\Sigma}}_b^\mu, \widetilde{\boldsymbol{\Sigma}}_b^\Sigma)$, respectively (more details in the following three paragraphs on $\mathcal{A}(\cdot)$, $\mathcal{S}(\cdot)$, and $\mathcal{F}(\cdot)$).

**Function $\mathcal{A}(\cdot)$: Statistics of the current Bin $b$'s Statistics.** As part of our probabilistic overall statistics, the *probabilistic overall mean* becomes a distribution with the mean (letting $\boldsymbol{\mu}_b = \bar{\mathbf{z}}$) and covariance (assuming diagonal covariance):

$$\boldsymbol{\mu}_b^\mu = \mathbb{E}[\bar{\mathbf{z}}] = \frac{1}{N_b} \sum_{i=1}^{N_b} \mathbf{z}_i^\mu, \qquad \boldsymbol{\mu}_b^\Sigma = \mathbb{V}[\bar{\mathbf{z}}] = \frac{1}{N_b^2} \sum_{i=1}^{N_b} \mathbf{z}_i^\Sigma.$$

Similarly, our *probabilistic overall covariance* becomes a matrix-variate distribution (Gupta & Nagar, 2018) with the mean:

$$\boldsymbol{\Sigma}_b^\mu = \frac{1}{N_b} \sum_{i=1}^{N_b} (\mathbf{z}_i - \bar{\mathbf{z}})^2 = \frac{1}{N_b} \sum_{i=1}^{N_b} \left[ \mathbf{z}_i^\Sigma + (\mathbf{z}_i^\mu)^2 - \left( [\boldsymbol{\mu}_b^\Sigma]_i + ([\boldsymbol{\mu}_b^\mu]_i)^2 \right) \right],$$

since $\mathbb{E}[\bar{\mathbf{z}}] = \boldsymbol{\mu}_b^\mu$ and $\mathbb{V}[\bar{\mathbf{z}}] = \boldsymbol{\mu}_b^\Sigma$. Note that the covariance of $\Sigma_b$, i.e., $\Sigma_b^\Sigma$, involves computing the fourth-order moments, which is computationally prohibitive. Therefore in practice, we directly set $\Sigma_b^\Sigma$ to zero for simplicity; empirically we observe that such simplified treatment already achieves promising performance improvement upon the state of the art.

**Function $\mathcal{S}(\cdot)$: Neighboring Data and Smoothed Statistics.** Next, we can borrow data with neighboring labels (from neighboring label bins) to compute the smoothed statistics of the current bin $b$ by applying a symmetric kernel $k(\cdot, \cdot)$ (e.g., Gaussian, Laplacian, and Triangular kernels). Specifically, the *probabilistic smoothed mean and covariance* are (assuming diagonal covariance):

$$\widetilde{\boldsymbol{\mu}}_b^\mu = \sum_{b' \in \mathcal{B}} k(y_b, y_{b'}) \boldsymbol{\mu}_{b'}^\mu, \quad \widetilde{\boldsymbol{\mu}}_b^\Sigma = \sum_{b' \in \mathcal{B}} k^2(y_b, y_{b'}) \boldsymbol{\mu}_{b'}^\Sigma, \quad \widetilde{\boldsymbol{\Sigma}}_b^\mu = \sum_{b' \in \mathcal{B}} k(y_b, y_{b'}) \boldsymbol{\Sigma}_{b'}.$$

**Function $\mathcal{F}(\cdot)$: Probabilistic Whitening and Recoloring.** We develop a probabilistic version of the whitening and re-coloring procedure (Sun et al., 2016) used in (Yang et al., 2021). Specifically, we produce the final probabilistic representation $\{\widetilde{\mathbf{z}}_i^\mu, \widetilde{\mathbf{z}}_i^\Sigma\}$ for each data point as:

$$\widetilde{\mathbf{z}}_i^\mu = (\mathbf{z}_i^\mu - \boldsymbol{\mu}_b^\mu) \cdot \sqrt{\frac{\widetilde{\boldsymbol{\Sigma}}_b^\mu}{\boldsymbol{\Sigma}_b^\mu}} + \widetilde{\boldsymbol{\mu}}_b^\mu, \quad \widetilde{\mathbf{z}}_i^\Sigma = (\mathbf{z}_i^\Sigma + \boldsymbol{\mu}_b^\Sigma) \cdot \sqrt{\frac{\widetilde{\boldsymbol{\Sigma}}_b^\mu}{\boldsymbol{\Sigma}_b^\mu}} + \widetilde{\boldsymbol{\mu}}_b^\Sigma. \tag{5}$$

Inspired by (Yang et al., 2021), we keep updating the probabilistic overall statistics, $\{\boldsymbol{\mu}_b^\mu, \boldsymbol{\mu}_b^\Sigma, \boldsymbol{\Sigma}_b\}$, and the probabilistic smoothed statistics, $\{\widetilde{\boldsymbol{\mu}}_b^\mu, \widetilde{\boldsymbol{\mu}}_b^\Sigma\}$, cross different epochs. The probabilistic representation $\{\widetilde{\mathbf{z}}_i^\mu, \widetilde{\mathbf{z}}_i^\Sigma\}$ are then re-parameterized (Kingma & Welling, 2014) into the final representation $\mathbf{z}_i$, and passed into the final layer (discussed in Sec. 3.4) to generate the prediction and uncertainty estimation. Note that the computation of statistics from multiple $\mathbf{x}$'s is only needed during training. During testing, VIR directly uses these statistics and therefore does not need to re-compute them.

## 3.4 CONSTRUCTING $p(y_i|\mathbf{z}_i)$

Our VIR's predictor $p(y_i|\mathbf{z}_i) \triangleq \mathcal{N}(y_i; \widehat{y}_i, \widehat{s}_i)$ predicts both the mean and variance for $y_i$ by first predicting the NIG distribution and then marginalizing out the latent variables. It is motivated by the following observations on label distribution smoothing (LDS) in (Yang et al., 2021) and deep evidential regression (DER) in (Amini et al., 2020), as well as intuitions on effective counts in conjugate distributions.

**LDS's Limitations in Our Probabilistic Imbalanced Regression Setting.** The motivation of LDS (Yang et al., 2021) is that the empirical label distribution can not reflect the real label distribution in an imbalanced dataset with a continuous label space; consequently, reweighting methods for imbalanced regression fail due to these inaccurate label densities. By applying a smoothing kernel on the empirical label distribution, LDS tries to recover the effective label distribution, with which reweighting methods can obtain 'better' weights to improve imbalanced regression. However, in our probabilistic imbalanced regression, one needs to consider both (1) the performance for the data with minority labels and (2) uncertainty estimation for each model. However, LDS only focuses on improving the accuracy, especially for the data with minority labels, and therefore does not provide uncertainty estimation, which is crucial to assess the predictions' reliability.

**DER's limitations in Our Probabilistic Imbalanced Regression Setting.** In DER (Amini et al., 2020), the predicted labels with their correspond uncertainties are produced by the representation of the posterior parameters in Normal Inverse Gamma (NIG) distribution $NIG(\gamma, \nu, \alpha, \beta)$, while the model is trained via minimizing the negative log-likelihood (NLL) of a Student-t distribution:

$$\mathcal{L}_i^{DER} = \frac{1}{2} \log(\frac{\pi}{\nu}) + (\alpha + \frac{1}{2}) \log((y_i - \gamma)^2 \nu + \Omega) - \alpha \log(\Omega) + \log(\frac{\Gamma(\alpha)}{\Gamma(\alpha + \frac{1}{2})}), \tag{6}$$

where $\Omega = 2\beta(1 + \nu)$. It is therefore nontrivial to properly incorporate a reweighting mechanism into the NLL. One straightforward approach is to directly reweight $\mathcal{L}_i^{DER}$ for different data points $(x_i, y_i)$. However, this contradicts the formulation of NIG and often leads to poor performance, as we verify in Sec. 4.

**Intuition of Pseudo-Counts for VIR.** To properly incorporate different reweighting methods, our VIR relies on the intuition of pseudo-counts (pseudo-observations) in conjugate distributions (Bishop, 2006). Assuming Gaussian likelihood, the *conjugate distributions* would be an NIG distribution (Bishop, 2006), i.e., $(\mu, \Sigma) \sim NIG(\gamma, \nu, \alpha, \beta)$, which means:

$$\mu \sim \mathcal{N}(\gamma, \Sigma/\nu), \quad \Sigma \sim \Gamma^{-1}(\alpha, \beta),$$

where $\Gamma^{-1}(\alpha, \beta)$ is an inverse gamma distribution. With a NIG prior distribution $NIG(\gamma_0, \nu_0, \alpha_0, \beta_0)$, the posterior distribution of the NIG after observing $n$ real data points are:

$$\gamma_n = \frac{\gamma_0 \nu_0 + n\Psi}{\nu_n}, \quad \nu_n = \nu_0 + n, \quad \alpha_n = \alpha_0 + \frac{n}{2}, \quad \beta_n = \beta_0 + \frac{1}{2}(\gamma_0^2 \nu_0) + \Phi, \tag{7}$$

where $\Psi = \bar{\mathbf{x}}$ and $\Phi = \frac{1}{2}(\sum_i \mathbf{x}_i^2 - \gamma_n^2 \nu_n)$. Here $\nu_0$ and $\alpha_0$ can be interpreted as virtual observations, i.e., *pseudo-counts or pseudo-observations* that contribute to the posterior distribution. Overall, the mean of posterior distribution above can be interpreted as an estimation from $(2\alpha_0 + n)$ observations, with $2\alpha_0$ virtual observations and $n$ real observations. Similarly, the variance can be interpreted an estimation from $(\nu + n)$ observations. This intuition is crucial in developing the predictor of our VIR.

**From Pseudo-Counts to Balanced Predictive Distributions.** Based on the intuition above, we construct our predictor (i.e., $p(y_i|\mathbf{z}_i)$) by (1) generating the parameters in the posterior distribution of NIG, (2) computing re-weighted parameters by imposing the importance weights obtained from LDS, and (3) producing the final prediction with corresponding uncertainty estimation.

Based on Eqn. 7, we feed the final representation $\{\mathbf{z}_i\}_{i=1}^N$ generated from the Sec. 3.3 (Eqn. 5) into a linear layer to output the intermediate parameters $n_i, \Psi_i, \Phi_i$ for data point $(\mathbf{x}_i, y_i)$:

$$n_i, \Psi_i, \Phi_i = \mathcal{G}(\mathbf{z}_i), \quad \mathbf{z}_i \sim q(\mathbf{z}_i|\{\mathbf{x}_i\}_{i=1}^N) = \mathcal{N}(\mathbf{z}_i; \widetilde{\mathbf{z}}_i^\mu, \widetilde{\mathbf{z}}_i^\Sigma)$$

We then apply the importance weights $\sum_{b' \in \mathcal{B}} k(y_b, y_{b'}))^{-\frac{1}{2}}$ calculated from the smoothed label distribution to the *pseudo-count* $n_i$ to produce the re-weighted parameters of posterior distribution of NIG. Along with the pre-defined prior parameters $(\gamma_0, \nu_0, \alpha_0, \beta_0)$, we are able to compute the parameters of posterior distribution $NIG(\gamma_i, \nu_i, \alpha_i, \beta_i)$ for $(\mathbf{x}_i, y_i)$:

$$\gamma_i^* = \frac{\gamma_0 \nu_0 + \left(\sum_{b' \in \mathcal{B}} k(y_b, y_{b'})\right)^{-\frac{1}{2}} \cdot n_i \Psi_i}{\nu_n^*}, \quad \nu_i^* = \nu_0 + \left(\sum_{b' \in \mathcal{B}} k(y_b, y_{b'})\right)^{-\frac{1}{2}} \cdot n_i,$$

$$\alpha_i^* = \alpha_0 + \left(\sum_{b' \in \mathcal{B}} k(y_b, y_{b'})\right)^{-\frac{1}{2}} \cdot \frac{n_i}{2}, \quad \beta_i^* = \beta_0 + \frac{1}{2}(\gamma_0^2 \nu_0) + \Phi_i.$$

Based on the NIG posterior distribution, we can then compute final prediction and uncertainty estimation as

$$\widehat{y}_i = \gamma_i^*, \quad \widehat{s}_i = \frac{\beta_i^*}{\nu_i^*(\alpha_i^* - 1)}.$$

We use an objective function similar to Eqn. 6, but with different definitions of $(\gamma, \nu, \alpha, \beta)$, to optimize our VIR model:

$$\mathcal{L}_i^{\mathcal{P}} = \mathbb{E}_{q_\phi(\mathbf{z}_i|\{\mathbf{x}_i\}_{i=1}^N)}\left[\frac{1}{2}\log(\frac{\pi}{\nu_i^*}) + (\alpha_i^* + \frac{1}{2})\log((y_i - \gamma_i^*)^2 \nu_n^* + \Omega) - \alpha_i^* \log(\omega_i^*) + \log(\frac{\Gamma(\alpha_i^*)}{\Gamma(\alpha_i^* + \frac{1}{2})})\right], \quad (8)$$

where $\omega_i^* = 2\beta_i^*(1 + \nu_i^*)$. Note that $\mathcal{L}_i^{\mathcal{P}}$ is part of the ELBO in Eqn. 1. Similar to (Amini et al., 2020), we use an additional regularization term to achieve better accuracy[1]:

$$\mathcal{L}_i^{\mathcal{R}} = (\nu + 2\alpha) \cdot |y_i - \widehat{y}_i|.$$

$\mathcal{L}_i^{\mathcal{P}}$ and $\mathcal{L}_i^{\mathcal{R}}$ together constitute the objective function for learning the predictor $p(\mathbf{y}_i|\mathbf{z}_i)$.

## 3.5 FINAL OBJECTIVE FUNCTION

Putting together Sec. 3.3 and Sec. 3.4, our final objective function (to minimize) for VIR is:

$$\mathcal{L}^{\mathcal{VIR}} = \sum_{i=1}^N \mathcal{L}_i^{\mathcal{VIR}}, \quad \mathcal{L}_i^{\mathcal{VIR}} = \lambda \mathcal{L}_i^{\mathcal{R}} - \mathcal{L}(\theta, \phi; \mathbf{x}_i, y_i) = \lambda \mathcal{L}_i^{\mathcal{R}} - \mathcal{L}_i^{\mathcal{P}} - \mathcal{L}_i^{\mathcal{D}} + \mathcal{L}_i^{\mathcal{KL}},$$

where $\mathcal{L}(\theta, \phi; \mathbf{x}_i, y_i) = \mathcal{L}_i^{\mathcal{P}} + \mathcal{L}_i^{\mathcal{D}} - \mathcal{L}_i^{\mathcal{KL}}$ is the ELBO in Eqn. 1. $\lambda$ adjusts the importance of the additional regularizer and the ELBO, and thus lead to a better result both on accuracy and uncertainty estimation.

## 3.6 DISCUSSION ON I.I.D. AND N.I.D. ASSUMPTIONS

**Generalization Error, Bias, and Variance.** We could analyze the generalization error of our VIR by bounding the generalization with the sum of three terms: (a) the bias of our estimator, (2) the variance of our estimator, (3) model complexity. Essentially VIR uses the N.I.D. assumption increases our estimator's bias, but significantly reduces its variance in the imbalanced setting. Since the model complexity is kept the same (using the same backbone neural network) as the baselines, N.I.D. will lead to a lower generalization error (see more discussion in Sec. A of the Appendix).

---

[1]Note that in DER, the total evidence $\Phi$ has a value $2\nu + \alpha$, but to the best of our knowledge, it would be more reasonable to use $\nu + 2\alpha$ as the total evidence for an NIG distribution (Bishop, 2006).

## 4 RESULTS

**Datasets.** In this work, we evaluate our methods in terms of prediction accuracy and uncertainty estimation on two imbalanced datasets[2], AgeDB (Moschoglou et al., 2017), IMDB-WIKI (Rothe et al., 2018). We follow the preprocessing procedures in DIR (Yang et al., 2021). Details for label density distributions and levels of imbalance are discussed in DIR (Yang et al., 2021).

*AgeDB-DIR*: We use AgeDB-DIR constructed in DIR (Yang et al., 2021), which contains 12.2K images for training and 2.1K images for validation and testing. The maximum age in this dataset is 101 and the minimum age is 0, and the number of images per bin varies between 1 and 353.

*IMDB-WIKI-DIR*: We use IMDB-WIKI-DIR constructed in DIR (Yang et al., 2021), which contains 191.5K training images and 11.0K validation and testing images. The maximum age is 186 and minimum age is 0; the maximum bin density is 7149, and minimum bin density is 1.

*STS-B-DIR*: We use STS-B-DIR constructed in DIR (Yang et al., 2021), which contains 5.2K pairs of training sentences and 1.0K pairs for validation and testing. This dataset is a collection of sentence pairs generated from news headlines, video captions, etc. Each pair is annotated by multiple annotators with a similarity score between 0 and 5.

**Baselines.** We use ResNet-50 (He et al., 2016) as our backbone network, and we describe the baselines below.

*Vanilla*: We use the term **VANILLA** to denote a plain model without adding any approaches.

*Synthetic-Sample-Based Methods*: Various existing imbalanced regression methods are also included as baselines; these include SMOTER (Torgo et al., 2013) and SMOGN (Branco et al., 2017). Furthermore, following DIR (Yang et al., 2021), in IMDB-WIKI-DIR, we also include another two methods: MIXUP (Zhang et al., 2018) and M-MIXUP (Verma et al., 2019).

*Cost-Sensitive Reweighting*: As shown in DIR (Yang et al., 2021), the square-root weighting variant (SQINV) baseline (i.e. $\left( \sum_{b' \in \mathcal{B}} k(y_b, y_{b'}) \right)^{-\frac{1}{2}}$) always outperforms Vanilla. Therefore, for simplicity and fair comparison, all our experiments (for both baselines and VIR) use SQINV weighting. To use SQINV in VIR, one simply needs to use the symmetric kernel $k(\cdot, \cdot)$ described in Sec. 3.3. To use SQINV in DER, we replace the final layer in DIR (Yang et al., 2021) with the DER layer (Amini et al., 2020) to produce the predictive distributions.

**Evaluation Metrics - Accuracy.** We follow the evaluation metrics in (Yang et al., 2021) to evaluate the accuracy of our proposed methods; these include Mean Absolute Error (MAE), Mean Squared Error (MSE), and Geometric Mean (GM). The formulas for these metrics are as follows:

$$\texttt{MAE} = \frac{1}{N} \sum_{i=1}^{N} |y_i - \widehat{y}_i|, \quad \texttt{MSE} = \frac{1}{N} \sum_{i=1}^{N} (y_i - \widehat{y}_i)^2, \quad \texttt{GM} = \left[ \prod_{i=1}^{N} |y_i - \widehat{y}_i| \right]^{\frac{1}{N}}.$$

**Evaluation Metrics - Uncertainty Estimation.** We use typical evaluation metrics for uncertainty estimation in regression problems to evaluate our produced uncertainty estimation; these include Negative Log Likelihood (NLL), Area Under Sparsification Error (AUSE). Eqn. 8 shows the formula for NLL, and more details regarding to AUSE can be found in (Ilg et al., 2018).

**Evaluation Process.** Following (Liu et al., 2019; Yang et al., 2021), for a data sample $x_i$ with its label $y_i$ which falls into the target bins $b_i$, we divide the label space into three disjoint subsets: many-shot region $\{b_i \in \mathcal{B} \mid y_i \in b_i \ \& \ |y_i| > 100\}$, medium-shot region $\{b_i \in \mathcal{B} \mid y_i \in b_i \ \& \ 20 \le |y_i| \le 100\}$, and few-shot region $\{b_i \in \mathcal{B} \mid y_i \in b_i \ \& \ |y_i| < 20\}$, where $|\cdot|$ denotes the cardinality of the set. We report results on the overall test set and these subsets with the accuracy metrics discussed above.

**Implementation Details.** We use ResNet-50 (He et al., 2016) for all experiments in AgeDB-DIR and IMDB-WIKI-DIR. We use the Adam optimizer (Kingma & Ba, 2015) to train all models for 100 epochs, with same learning rate and decay by 0.1 and the 60-th and 90-th epoch, respectively. In order to determine the optimal batch size for training, we try different batch sizes and achieve the same

---

[2]Among the five datasets proposed in (Yang et al., 2021), only four of them are publicly available. In this paper we use the largest (IMDB-WIKI) and the smallest (AgeDB) among the four to evaluate our method.

Table 1: Evaluation results of accuracy on AgeDB-DIR.

| Metrics | MSE↓ | | | | MAE↓ | | | | GM↓ | | | |
|---|---|---|---|---|---|---|---|---|---|---|---|---|
| Shot | All | Many | Med. | Few | All | Many | Med. | Few | All | Many | Med. | Few |
| VANILLA (Yang et al., 2021) | 101.28 | 78.40 | 131.17 | 256.32 | 7.79 | 6.70 | 9.42 | 13.98 | 5.18 | 4.53 | 6.75 | 11.54 |
| DEEP ENSEMBLE (Lakshminarayanan et al., 2017) | 100.94 | 79.30 | 129.95 | 249.18 | 7.73 | 6.62 | 9.37 | 13.90 | 4.87 | 4.37 | 6.50 | 11.35 |
| SMOTER (Torgo et al., 2013) | 114.34 | 93.35 | 129.89 | 244.57 | 8.16 | 7.39 | 8.65 | 12.28 | 5.21 | 4.65 | 5.69 | 8.49 |
| SMOGN (Branco et al., 2017) | 117.29 | 101.36 | 133.86 | 232.90 | 8.26 | 7.64 | 9.01 | 12.09 | 5.36 | 4.90 | 6.19 | 8.44 |
| SQINV (Yang et al., 2021) | 104.76 | 92.67 | 127.04 | 205.16 | 7.92 | 7.42 | 8.80 | 11.46 | 5.03 | 4.81 | 5.72 | 8.23 |
| DER (Amini et al., 2020) | 106.81 | 91.32 | 122.45 | 209.76 | 8.11 | 7.36 | 9.03 | 12.69 | 5.31 | 4.65 | 6.48 | 10.52 |
| FDS (Yang et al., 2021) | 109.78 | 93.99 | 124.96 | 216.97 | 8.12 | 7.52 | 8.68 | 12.25 | 5.13 | 4.80 | 5.97 | 8.85 |
| LDS (Yang et al., 2021) | 102.22 | 83.62 | 128.73 | 204.64 | 7.67 | 6.98 | 8.86 | 10.89 | 4.85 | 4.39 | 5.80 | 7.45 |
| LDS + FDS (Yang et al., 2021) | 102.16 | 86.99 | 128.04 | 199.18 | 7.82 | 7.19 | 9.08 | 11.24 | 5.01 | 4.56 | 6.10 | 7.02 |
| FDS + RANKSIM (Gong et al., 2022) | 83.51 | 71.99 | 99.14 | 149.05 | 7.02 | 6.49 | 7.84 | 9.68 | 4.53 | 4.13 | 5.37 | 6.89 |
| LDS + FDS + RANKSIM (Gong et al., 2022) | 84.96 | 74.27 | 93.64 | 161.92 | 7.03 | 6.54 | 7.68 | 9.92 | 4.45 | 4.07 | 5.23 | 6.35 |
| LDS + FDS + DER (Yang et al., 2021; Amini et al., 2020) | 112.62 | 94.21 | 140.03 | 210.72 | 8.18 | 7.44 | 9.52 | 11.45 | 5.30 | 4.75 | 6.74 | 7.68 |
| **VIR (OURS)** | **86.89** | **77.69** | **96.55** | **145.76** | **7.14** | **6.67** | **7.70** | **9.52** | **4.58** | **4.27** | **5.09** | **6.31** |
| OURS VS. VANILLA | +14.39 | +0.71 | +34.62 | +110.56 | +0.65 | +0.03 | +1.72 | +4.46 | +0.60 | +0.26 | +1.66 | +5.23 |
| OURS VS. SQINV | +17.87 | +14.98 | +30.49 | +59.40 | +0.78 | +0.75 | +1.10 | +1.94 | +0.45 | +0.54 | +0.63 | +1.92 |
| OURS VS. DER | +19.92 | +13.63 | +25.90 | +64.00 | +0.97 | +0.69 | +1.33 | +3.17 | +0.73 | +0.38 | +1.39 | +4.21 |
| OURS VS. LDS + FDS (SOTA IN DIR) | +15.27 | +9.30 | +31.49 | +53.42 | +0.68 | +0.52 | +1.38 | +1.72 | +0.43 | +0.29 | +1.01 | +0.71 |

conclusion as the DIR paper, i.e., the optimal batch size is 256 when other hyperparameters are fixed. Therefore, we stick to the batch size of 256 through out the experiments in the paper. Meanwhile, we use the same hyperparameters as in DIR (Yang et al., 2021).

We use PyTorch to implement our method. For fair comparison, we implemented a PyTorch version for the official TensorFlow implementation of DER(Amini et al., 2020). To make sure we can obtain the reasonable uncertainty estimations, we restrict the range for $\alpha$ to $[1.5, \infty)$ instead of $[1.0, \infty)$ in DER. Besides, in the activation function *SoftPlus*, we set the hyperparameter *beta* to 0.1. As discussed in Sec. 3.4, we implement a layer which produces the parameters $n, \Psi, \Omega$. We assign 2 as the minimum number for $n$, and use the same hyperparameter settings for activation function for DER layer.

To search for a combination hyperparameters of prior distribution $\{\gamma_0, \nu_0, \alpha_0, \beta_0\}$ for NIG, we combine grid search method and random search method (Bergstra & Bengio, 2012) to select the best hyperparameters. We first intuitively assign a value and a proper range with some step sizes which correspond to the hyperparameters, then, we apply grid search to search for the best combination for the hyperparameters on prior distributions. After locating a smaller range for each hyperparameters, we use random search to search for better combinations, if it exists. In the end, we find our best hyperparameter combinations for NIG prior distributions.

## 4.1 RESULTS FOR IMBALANCED REGRESSION ACCURACY

We report the accuracy of different methods in Table 1 and Table 2 for AgeDB-DIR and IMDB-WIKI-DIR, respectively[3]. In both tables, we can conclude that our methods outperform the baselines in their categories. For ablation studies, see Table 5 and Table 6 of the Appendix. Note that to ensure fair and solid comparison, we re-run the DIR methods based on our machine and software settings[4].

**Overall Performance.** As shown in the last category (i.e., last four rows) of both tables, our proposed method's best variants compare favorably against the state of the art including DIR variants (Yang et al., 2021) and DER (Amini et al., 2020), especially on the imbalanced data samples (i.e., in the few-shot columns). This verifies the effectiveness of our methods in terms of overall performance.

## 4.2 RESULTS FOR IMBALANCED REGRESSION UNCERTAINTY ESTIMATION

Different from DIR (Yang et al., 2021) which only focuses on accuracy, we create a new benchmark for uncertainty estimation in imbalanced regression. Table 3 and Table 4 show the results on uncertainty estimation for two datasets AgeDB-DIR and IMDB-WIKI-DIR, respectively. Note that most baselines from Table 1 and Table 2 are *deterministic* methods (as opposed to probabilistic

---

[3]Results for STS-B-DIR are reported in Table 7, Table 8, and Table 9 of the Appendix.

[4]We find that due to differences in PyTorch, GPU, and CUDA versions, as well as numbers of GPUs used for parallel training, the results in DIR may vary. Furthermore, the randomness in multiple workers in the Dataloader also affect the performance.

Table 2: Evaluation results of accuracy on IMDB-WIKI-DIR.

| Metrics | MSE ↓ | | | | MAE ↓ | | | | GM ↓ | | | |
|---|---|---|---|---|---|---|---|---|---|---|---|---|
| Shot | All | Many | Med. | Few | All | Many | Med. | Few | All | Many | Med. | Few |
| VANILLA (Yang et al., 2021) | 135.48 | 107.01 | 352.02 | 973.73 | 7.99 | 7.18 | 14.88 | 26.72 | 4.51 | 4.12 | 10.46 | 21.40 |
| MIXUP (Zhang et al., 2018) | 141.11 | 109.13 | 389.95 | 1037.98 | 8.22 | 7.29 | 16.23 | 28.11 | 4.68 | 4.22 | 12.28 | 23.55 |
| M-MIXUP (Verma et al., 2019) | 137.45 | 108.33 | 363.72 | 957.53 | 8.22 | 7.39 | 15.24 | 26.70 | 4.80 | 4.39 | 10.85 | 21.86 |
| SMOTER (Torgo et al., 2013) | 138.75 | 111.55 | 346.09 | 935.89 | 8.14 | 7.42 | 14.15 | 25.28 | 4.64 | 4.30 | 9.05 | 19.46 |
| SMOGN (Branco et al., 2017) | 136.09 | 109.15 | 339.09 | 944.20 | 8.03 | 7.30 | 14.02 | 25.93 | 4.63 | 4.30 | 8.74 | 20.12 |
| SQINV (Yang et al., 2021) | 134.36 | 111.23 | 308.63 | 834.08 | 7.87 | 7.24 | 12.44 | 22.76 | 4.47 | 4.22 | 7.25 | 15.10 |
| DER (Amini et al., 2020) | 133.81 | 107.51 | 332.90 | 916.18 | 7.85 | 7.18 | 13.35 | 24.12 | 4.47 | 4.18 | 8.18 | 15.18 |
| FDS (Yang et al., 2021) | 131.93 | 107.76 | 311.29 | 880.32 | 7.80 | 7.20 | 12.64 | 23.20 | 4.39 | 4.16 | 7.04 | 13.42 |
| LDS (Yang et al., 2021) | 133.93 | 109.70 | 320.26 | 830.81 | 7.91 | 7.30 | 13.02 | 22.41 | 4.48 | 4.22 | 7.72 | 13.75 |
| LDS + FDS (Yang et al., 2021) | 136.72 | 112.76 | 322.50 | 811.83 | 8.08 | 7.47 | 13.21 | 22.54 | 4.66 | 4.39 | 8.01 | 14.33 |
| LDS + FDS + DER (Yang et al., 2021; Amini et al., 2020) | 120.86 | **97.75** | **297.64** | 873.10 | 7.24 | **6.64** | **11.87** | 23.44 | 3.93 | 3.69 | 6.64 | 16.00 |
| **VIR (OURS)** | **119.60** | 99.25 | 298.85 | **809.34** | **7.23** | 6.66 | 11.90 | **21.78** | **3.90** | **3.68** | **6.51** | **13.34** |
| **OURS** VS. VANILLA | +15.88 | +7.76 | +53.17 | +164.39 | +0.76 | +0.52 | +2.98 | +4.94 | +0.61 | +0.44 | +3.95 | +8.06 |
| **OURS** VS. SQINV | +14.76 | +11.98 | +9.78 | +24.74 | +0.64 | +0.58 | +0.54 | +0.98 | +0.57 | +0.54 | +0.74 | +1.76 |
| **OURS** VS. DER | +14.21 | +8.26 | +34.05 | +106.84 | +0.62 | +0.52 | +1.45 | +2.34 | +0.57 | +0.50 | +1.67 | +1.84 |
| **OURS** VS. LDS + FDS (SOTA IN DIR) | +17.12 | +13.51 | +23.65 | +2.49 | +0.85 | +0.81 | +1.31 | +0.76 | +0.76 | +0.71 | +1.50 | +0.99 |

methods like ours) and *cannot provide uncertainty estimation*; therefore they are not applicable here. To show the superiority of our VIR model, we create a strongest baseline by concatenating the DIR variants (LDS + FDS) with the DER (Amini et al., 2020).

Results show that VIR outperform the baselines in all few-shot metrics. In some categories, VIR may not perform better in the overall, many-shot and median shot metrics, but the gap tends to be minimal. Note that our proposed methods mainly focus on the imbalanced setting, therefore we also focus on the few-shot metrics. Lastly, comparing our model variant with the best performance against the baseline (DER), we can conclude that our methods successfully improve uncertainty estimation in the probabilistic imbalanced regression setting.

Table 3: Uncertainty estimation results on AgeDB-DIR.

| Metrics | NLL ↓ | | | | AUSE ↓ | | | |
|---|---|---|---|---|---|---|---|---|
| Shot | All | Many | Med. | Few | All | Many | Med. | Few |
| DEEP ENSEMBLE (Lakshminarayanan et al., 2017) | 5.311 | 4.031 | 6.726 | 8.523 | 0.541 | 0.626 | 0.466 | 0.483 |
| DER (Amini et al., 2020) | 3.936 | 3.768 | 3.865 | 4.421 | 0.590 | 0.449 | 0.468 | 0.500 |
| LDS + FDS + DER (Yang et al., 2021; Amini et al., 2020) | 3.794 | 3.699 | 3.969 | 4.214 | 0.463 | **0.260** | 0.392 | 0.617 |
| **VIR (OURS)** | **3.703** | **3.598** | 3.805 | **4.196** | **0.437** | 0.474 | **0.319** | **0.413** |
| **OURS** VS. DER | +0.064 | +0.071 | +0.060 | +0.225 | +0.153 | +0.026 | +0.007 | +0.036 |

We also observe that the improvements of the uncertainty estimation on IMDB-WIKI are larger than those on Age-DB. We suspect that this because IMDB-WIKI contains much more training, validating and testing data, therefore enjoying more stable

Table 4: Uncertainty estimation results on IMDB-WIKI-DIR.

| Metrics | NLL ↓ | | | | AUSE ↓ | | | |
|---|---|---|---|---|---|---|---|---|
| Shot | All | Many | Med. | Few | All | Many | Med. | Few |
| DER (Amini et al., 2020) | 3.850 | 3.699 | 4.997 | 6.638 | 0.813 | 0.802 | 0.650 | 0.541 |
| LDS + FDS + DER (Yang et al., 2021; Amini et al., 2020) | 3.683 | 3.602 | **4.391** | 5.697 | 0.784 | 0.670 | **0.455** | 0.483 |
| **VIR (OURS)** | **3.652** | **3.568** | 4.419 | **5.560** | **0.622** | **0.645** | 0.511 | **0.374** |
| **OURS** VS. DER | +0.198 | +0.131 | +0.578 | +1.078 | +0.191 | +0.157 | +0.202 | +0.167 |

uncertainty estimation improvements brought by VIR compared to those in Age-DB.

## 5 CONCLUSION

We identify the problem of probabilistic deep imbalanced regression, which aims to both improve accuracy and obtain reasonable uncertainty estimation in imbalanced regression. We propose VIR, which can use any deep regression models as backbone networks. VIR borrows data with similar regression labels to produce the probabilistic representations and modulates the conjugate distributions to impose probabilistic reweighting on imbalanced data. Furthermore, we create new benchmarks for uncertainty estimation on imbalanced regression. Experiments show that our methods outperform state-of-the-art imbalanced regression models in terms of both accuracy and uncertainty estimation. Future work may include (1) improving VIR by better approximating *variance of the variances* in probability distributions, and (2) developing novel approaches that can achieve stable performance even on imbalanced data with limited sample size, and (3) exploring techniques such as mixture density networks (Bishop, 1994) to enable multi-modality in the latent distribution, thereby further improving the performance.

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

## A    DISCUSSION ON I.I.D. AND N.I.D. ASSUMPTIONS

**Generalization Error, Bias, and Variance.** We could analyze the generalization error of our VIR by bounding the generalization with the sum of three terms: (a) the bias of our estimator, (2) the variance of our estimator, (3) model complexity. Essentially VIR uses the N.I.D. assumption increases our estimator's bias, but significantly reduces its variance in the imbalanced setting. Since the model complexity is kept the same (using the same backbone neural network) as the baselines, N.I.D. will lead to a lower generalization error.

**Variance of Estimators in Imbalanced Settings.** In the imbalanced setting, one typically use inverse weighting to produced an unbiased estimator (i.e., making the first term of the aforementioned bound zero). However, for data with extremely low density, its inverse would be extremely large, therefore leading to a very large variance for the estimator. Our VIR replaces I.I.D. with N.I.D. to "smooth out" such singularity, and therefore significantly lowers the variance of the estimator (i.e., making the second term of the aforementioned bound smaller), and ultimately lowers the generalization error.

## B    ADDITIONAL EXPERIMENT RESULTS

### B.1    ABLATION STUDY ON VIR

In this section, we include ablation studies to verify that our VIR can outperform its counterparts in DIR (i.e., smoothing on the latent space) and DER (i.e., NIG distribution layers).

**Ablation Study on** $q(\mathbf{z}_i|\{\mathbf{x}_i\}_{i=1}^N)$**.** To verify the effectiveness of VIR's encoder $q(\mathbf{z}_i|\{\mathbf{x}_i\}_{i=1}^N)$, we replace VIR's predictor $p(y_i|\mathbf{z}_i)$ with a linear layer (as in DIR). Table 5 shows that compared to its counterpart, FDS (Yang et al., 2021), our encoder-only VIR still leads to a considerable improvements even without generating the NIG distribution, therefore verifying the effectiveness of our VIR's $q(\mathbf{z}_i|\{\mathbf{x}_i\}_{i=1}^N)$.

Table 5: Ablation study on AgeDB-DIR in terms of accuracy.

| Metrics | MSE ↓ | | | | MAE ↓ | | | |
|---|---|---|---|---|---|---|---|---|
| Shot | All | Many | Med. | Few | All | Many | Med. | Few |
| FDS (Yang et al., 2021) | 109.78 | 93.99 | 124.96 | 216.97 | 8.12 | 7.52 | **8.68** | 12.25 |
| **ENCODER-ONLY VIR (OURS)** | 95.99 | 81.89 | 121.78 | 157.92 | 7.57 | 6.97 | 8.72 | **10.03** |
| DER (Amini et al., 2020) | 106.81 | 91.32 | 122.45 | 209.76 | 8.11 | 7.36 | 9.03 | 12.69 |
| **PREDICTOR-ONLY VIR (OURS)** | 88.96 | 74.79 | 95.85 | 203.76 | 7.28 | 6.68 | 7.76 | 11.63 |

**Ablation Study on** $p(y_i|\mathbf{z}_i)$**.** To verify the effectiveness of VIR's predictor $p(y_i|\mathbf{z}_i)$, we replace VIR's encoder $q(\mathbf{z}_i|\{\mathbf{x}_i\}_{i=1}^N)$ with a simple deterministic encoder as in DER (Amini et al., 2020). Table 5 and Table 6 show that compared to DER, the counterpart of VIR's predictor, our VIR's predictor still outperforms than DER, demonstrating its effectiveness; this verifies our claim (Sec. 3.4) that directly reweighting DER breaks NIG and leads to poor performance.

Table 6: Ablation study on AgeDB-DIR in terms of uncertainty estimation.

| Metrics | NLL ↓ | | | | AUSE ↓ | | | |
|---|---|---|---|---|---|---|---|---|
| Shot | All | Many | Med. | Few | All | Many | Med. | Few |
| DER Amini et al. (2020) | 3.936 | 3.768 | 3.865 | 4.421 | 0.590 | 0.449 | 0.468 | 0.500 |
| **PREDICTOR-ONLY VIR (OURS)** | **3.887** | **3.755** | **3.854** | **4.394** | **0.443** | **0.387** | **0.390** | **0.407** |

### B.2    RESULT ON STS-B-DIR DATASET

In this section, we report the accuracy and uncertainty evaluation on STS-B-DIR (more details for the dataset is in DIR (Yang et al., 2021)). From Table 7, Table 8, and Table 9 below, we can conclude

Table 7: Evaluation results of accuracy on STS-B-DIR.

| Metrics | MSE ↓ | | | | MAE ↓ | | | | GM ↓ | | | |
|---|---|---|---|---|---|---|---|---|---|---|---|---|
| Shot | All | Many | Med. | Few | All | Many | Med. | Few | All | Many | Med. | Few |
| INV | 1.031 | 0.930 | 1.426 | 1.152 | 0.825 | 0.783 | 1.004 | 0.850 | 0.567 | 0.537 | 0.744 | 0.535 |
| DIR (Yang et al., 2021) | 1.000 | 0.912 | 1.368 | 1.055 | 0.812 | 0.772 | 0.989 | 0.809 | 0.560 | 0.535 | 0.739 | 0.477 |
| DIR + DER (Yang et al., 2021; Amini et al., 2020) | 1.007 | 0.880 | 1.535 | 1.086 | 0.812 | 0.757 | 1.046 | 0.842 | 0.558 | 0.518 | 0.765 | 0.574 |
| **VIR (OURS)** | **0.895** | **0.799** | **1.309** | **0.919** | **0.760** | **0.718** | **0.960** | **0.732** | **0.509** | **0.493** | **0.669** | **0.377** |

that our model also outperforms all baselines in terms of both accuracy metrics and uncertainty estimation metrics in this NLP dataset; this verifies the superiority of our model for NLP datasets.

Table 8: Evaluation results of accuracy on STS-B-DIR.

| Metrics | Pearson ↑ | | | | Spearman ↑ | | | |
|---|---|---|---|---|---|---|---|---|
| Shot | All | Many | Med. | Few | All | Many | Med. | Few |
| INV | 0.718 | 0.701 | 0.612 | 0.705 | 0.723 | 0.678 | 0.530 | 0.685 |
| DIR (YANG ET AL., 2021) | 0.732 | 0.711 | 0.646 | 0.742 | 0.731 | 0.672 | 0.519 | 0.739 |
| DIR + DER (YANG ET AL., 2021; AMINI ET AL., 2020) | 0.729 | 0.714 | 0.635 | 0.731 | 0.730 | 0.680 | 0.526 | 0.699 |
| **VIR (OURS)** | **0.765** | **0.740** | **0.663** | **0.770** | **0.770** | **0.713** | **0.534** | **0.770** |

Table 9: Uncertainty estimation results on STS-B-DIR.

| Metrics | NLL ↓ | | | | AUSE ↓ | | | |
|---|---|---|---|---|---|---|---|---|
| Shot | All | Many | Med. | Few | All | Many | Med. | Few |
| DIR + DER (YANG ET AL., 2021; AMINI ET AL., 2020) | 2.561 | 2.514 | 2.880 | 2.358 | 0.672 | 0.581 | 0.609 | 0.615 |
| **VIR (OURS)** | **1.996** | **1.810** | **2.754** | **2.152** | **0.591** | **0.575** | **0.602** | **0.510** |

### B.3 DIFFERENCE BETWEEN DIR'S AND OUR REPRODUCED RESULTS

To reproduce the results on AgeDB, we use exactly the same settings as in DIR's code (Yang et al., 2021) (i.e., by directly running their code on our machines without modifying hyperparameters). for each model in DIR we report, we use five different random seeds to produce five results. We then report the performance by taking the average of them. Table 10 and Table 11 show the example for SQINV and LDS+FDS on AgeDB-DIR. From the table we can see that under our hardware and

Table 10: Results of running SQINV for 5 different random seeds on AgeDB.

| Metrics | MSE ↓ | | | | MAE ↓ | | | | GM ↓ | | | |
|---|---|---|---|---|---|---|---|---|---|---|---|---|
| Shot | All | Many | Med. | Few | All | Many | Med. | Few | All | Many | Med. | Few |
| SQINV 1 | 107.02 | 90.71 | 131.5 | 193.39 | 8.04 | 7.40 | 9.01 | 11.33 | 5.15 | 4.73 | 8.81 | 8.22 |
| SQINV 2 | 111.55 | 93.43 | 141.03 | 209.17 | 8.12 | 7.47 | 9.17 | 11.58 | 5.21 | 4.85 | 5.75 | 8.25 |
| SQINV 3 | 114.33 | 96.83 | 134.56 | 223.86 | 8.21 | 7.59 | 9.01 | 11.81 | 5.17 | 4.74 | 5.85 | 8.27 |
| SQINV 4 | 106.24 | 91.81 | 120.26 | 203.78 | 7.94 | 7.39 | 8.58 | 11.39 | 5.06 | 4.74 | 5.41 | 7.66 |
| SQINV 5 | 104.73 | 90.24 | 127.33 | 208.05 | 7.99 | 7.47 | 8.98 | 11.49 | 5.07 | 4.79 | 5.68 | 7.98 |
| SQINV AVG | 108.77 | 92.60 | 130.94 | 207.65 | 8.06 | 7.46 | 8.95 | 11.52 | 5.13 | 4.77 | 6.30 | 8.08 |
| SQINV STD | 12.89 | 5.67 | 48.46 | 96.71 | 0.01 | 0.01 | 0.04 | 0.03 | 0.01 | 0.01 | 1.60 | 0.05 |
| SQINV RESULTS FROM (YANG ET AL., 2021) | 105.14 | 87.21 | 127.66 | 212.30 | 7.81 | 7.16 | 8.80 | 11.20 | 4.99 | 4.57 | 5.73 | 7.77 |

software environments, the SQINV model and LDS+FDS model (SOTA in DIR) could not perform as well as it is reported in DIR Yang et al. (2021), therefore for fair comparison, we use our replicated performance rather than theirs.

### B.4 ABLATION STUDY ON $\lambda$

In this section, we include ablation studies on the $\lambda$ in our objective function. For $\lambda \in \{10.0, 1.0, 0.1, 0.01, 0.001\}$, we run our VIR model on the AgeDB dataset. Table 12 shows the results. We can conclude that when $\lambda = 0.1$, our model achieves the best performance.

Table 11: Results of running LDS+FDS for 5 different random seeds on AgeDB.

| Metrics | MSE ↓ | | | | MAE ↓ | | | | GM ↓ | | | |
|---|---|---|---|---|---|---|---|---|---|---|---|---|
| Shot | All | Many | Med. | Few | All | Many | Med. | Few | All | Many | Med. | Few |
| LDS+FDS 1 | 104.33 | 88.67 | 128.99 | 194.06 | 7.87 | 7.26 | 8.97 | 10.88 | 5.02 | 4.60 | 5.87 | 7.51 |
| LDS+FDS 2 | 104.59 | 94.63 | 125.60 | 200.14 | 7.98 | 7.44 | 8.77 | 11.16 | 5.00 | 4.71 | 5.62 | 7.81 |
| LDS+FDS 3 | 110.17 | 95.97 | 123.24 | 208.11 | 8.07 | 7.54 | 8.71 | 11.41 | 5.09 | 4.73 | 5.71 | 7.48 |
| LDS+FDS 4 | 102.68 | 98.20 | 126.41 | 201.16 | 8.02 | 7.50 | 8.82 | 11.34 | 5.08 | 4.63 | 5.74 | 7.56 |
| LDS+FDS 5 | 105.77 | 91.07 | 127.00 | 185.85 | 7.93 | 7.35 | 8.80 | 10.96 | 5.07 | 4.74 | 5.52 | 7.73 |
| LDS+FDS AVG | 105.51 | 93.71 | 126.25 | 197.86 | 7.97 | 7.42 | 8.81 | 11.15 | 5.05 | 4.68 | 5.69 | 7.62 |
| LDS+FDS STD | 6.41 | 11.70 | 3.52 | 55.97 | 0.01 | 0.01 | 0.01 | 0.04 | 0.01 | 0.03 | 0.01 | 0.02 |
| LDS+FDS RESULTS FROM (YANG ET AL., 2021) | 99.46 | 84.10 | 112.20 | 209.27 | 7.55 | 7.01 | 8.24 | 10.79 | 4.72 | 4.36 | 5.45 | 6.79 |

Table 12: Ablation study on $\lambda$ for VIR on AgeDB-DIR

| Metrics | MSE ↓ | | | | MAE ↓ | | | | NLL ↓ | | | |
|---|---|---|---|---|---|---|---|---|---|---|---|---|
| Shot | All | Many | Med. | Few | All | Many | Med. | Few | All | Many | Med. | Few |
| $\lambda = 10.0$ | 104.31 | 91.01 | 116.43 | 196.35 | 7.88 | 7.38 | 8.42 | 11.13 | 3.827 | 3.733 | 4.140 | 4.407 |
| $\lambda = 1.0$ | 104.10 | 87.28 | 128.26 | 196.12 | 7.83 | 7.21 | 8.81 | 10.89 | 3.848 | 3.738 | 4.041 | 4.356 |
| $\lambda = 0.1$ | 86.28 | 76.87 | 101.57 | 132.90 | 7.19 | 6.75 | 7.97 | 9.19 | 3.785 | 3.694 | 3.963 | 4.151 |
| $\lambda = 0.01$ | 86.86 | 76.58 | 99.95 | 147.82 | 7.12 | 6.69 | 7.72 | 9.59 | 3.887 | 3.797 | 4.007 | 4.401 |
| $\lambda = 0.001$ | 87.25 | 74.13 | 104.78 | 162.64 | 7.13 | 6.64 | 7.92 | 9.63 | 3.980 | 3.868 | 4.161 | 4.546 |

