# OpenReview forum: "Variational Imbalanced Regression"
_ICLR.cc/2023/Conference — Submitted to ICLR 2023_

### Official Review · Reviewer_5JWG · 2022-10-21

**Confidence:** 4
**Correctness:** 4
**Technical Novelty And Significance:** 3
**Empirical Novelty And Significance:** 2
**Recommendation:** 5

**Clarity, Quality, Novelty And Reproducibility:**


Clarity:

[+] the paper is well-structured and easy to follow.

Novelty:

[+] it is essential for regression tasks to consider uncertainty estimation.

[-] the overall method is more like a combination of FDS [1] and DER [2] to me in the current presentation. It could be better motivated why probabilistic latent code is better than FDS to improve deep imbalanced regression.

Quality:

I am highly concerned about the experimental setup -- 1) fair comparison between VIR and other SOTAs. Reproducing worse SOTA results without error bars is not a good practice; 2) more datasets should be considered in order to claim the method as the contribution in general DIR; 3) no discussion or empirical comparison to the missing related work [3, 4].

[-] The SOTA results are much worse than the ones reported in the original paper [1]. For example, in Table 1, LDS+FDS: *MAE all* from original 7.55 to reported 7.82, *GM all* from original 4.72 to reported 5.01. Also in Table 2, LDS+FDS: *MAE all* from 7.78 to 8.08, *GM all* from 4.37 to 4.66. Did the authors reproduce all results? If the SOTA is not robust and the variance between different runs is high, please consider reporting the error bar for both methods (SOTA and VIR). It is not a good practice to replace the prior work results with single runs.

[-] I’m concerned about the conditional log-likelihood $\mathcal{L}^D$ in Eq. (1). Did the authors have any ablation study on the weighting parameter $\lambda$? The overall task is discriminative and evaluated on the accuracy of predicting the regression label. The weighting of this conditional log-likelihood term should be discussed. This term can dominate the objective since the authors use image-based age estimation datasets with high-dimensional $x$.


[1] Yang et al.,  Delving into Deep Imbalanced Regression. ICML 2021 \
[2] Amini et al., Deep Evidential Regression. NeurIPS 2020 \
[3] Ren et al., Balanced MSE for Imbalanced Visual Regression, CVPR 2022 \
[4] Gong et al., RankSim: Ranking Similarity Regularization for Deep Imbalanced Regression, ICML 2022 \


**Strength And Weaknesses:**

Strength:

(1) The authors introduce the setting of uncertainty estimation to the deep imbalanced regression. I appreciate this contribution since DIR was proposed very recently by Yang et al. (ICML 2021), and this can be a good contribution to the future investigation of DIR.

(2) The method is interesting, especially for the reweighting scheme of the balanced predictive distribution.

Weakness:

(1) Considering the novelty, I think VIR is more like a combination of [1] and [2] from the current manuscript. Though VIR is empirically superior and introduces uncertainty estimation as a byproduct, it is still vague how VIR is motivated compared to the FDS in [1], since both methods have kernel estimation as the core contribution. And the idea of extending deterministic space to a probabilistic one is interesting but lacks more motivation.

(2) Missing related work [3, 4]. Both were published before the submission deadline and the authors should discuss the difference and compare VIR with them.

(3) The experimental setup can be improved. First, the authors only choose two age estimation datasets whose distributions are similar. Second, the reported numbers are not very convincing (please see the next section for quality and reproducibility).

(4) I’m not fully convinced by the choice of Gaussian distribution as the prior distribution. For some specific datasets with regression labels like ages, this may possibly be a suitable assumption. But the authors claim the contributions to the task of general deep imbalanced regression, so I encourage the authors to conduct experiments on other types of regression datasets. It will strongly improve the contribution of this work if the authors can consider more datasets proposed in [1] or [3].

[1] Yang et al.,  Delving into Deep Imbalanced Regression. ICML 2021 \
[2] Amini et al., Deep Evidential Regression. NeurIPS 2020 \
[3] Ren et al., Balanced MSE for Imbalanced Visual Regression, CVPR 2022 \
[4] Gong et al., RankSim: Ranking Similarity Regularization for Deep Imbalanced Regression, ICML 2022 \




**Summary Of The Paper:**

The authors introduce a probabilistic model called Variational Imbalanced Regression (VIR) to deal with the problem of imbalanced regression. VIR borrows data with similar regression labels to compute the latent representation’s variational distribution. VIR first encodes a data point into a probabilistic representation and then mixes it with neighboring representations. The proposed method not only performs well in imbalanced regression but naturally produces reasonable uncertainty estimation as a byproduct.

**Summary Of The Review:**

I like the idea of extending conventional deterministic encoding to be probabilistic for deep imbalanced regression based on the formulation of variational autoencoders. The presentation of the idea is clear and technically sound. However, the experimental evaluation needs to be improved to make the method more convincing. Therefore, I recommend rejection at this moment but encourage the authors to improve and submit to future venues.

---

> ### Author Response · Authors · 2022-11-16
> **Thank you and response to your comments [1/4]**
>
> Thank you for your valuable comments. We are glad that you find our setting a ``"good contribution"`` and our method ``"interesting"``. Below we address your questions one by one.
>
> **Q1: "Considering the novelty, I think VIR is more like a combination of [1] and [2] from the current manuscript. Though VIR is empirically superior and introduces uncertainty estimation as a byproduct, it is still vague how VIR is motivated compared to the FDS in [1], since both methods have kernel estimation as the core contribution. And the idea of extending deterministic space to a probabilistic one is interesting but lacks more motivation."**
>
> This is a good question. We would like to clarify that our work is substantially different from [1], and our novelty rests on both proposing a new problem and proposing new methods to address the new problem. Specifically,
>
> **(1)** We identify the problem of probabilistic deep imbalanced regression as well as two desiderata, balanced accuracy and uncertainty estimation, for the problem.
>
> **(2)** We propose VIR to simultaneously cover these two desiderata and achieve state-of-the-art performance compared to existing methods.
>
> **(3)** Our VIR is a probabilistic generative model. Inspired from VAE, our VIR has encoder and decoder, and we introduce N.I.D. assumption and conjugate posterior distribution, which is a principle framework and not just a combination of DIR [1] and DER [2].
>
> **(4)** As a byproduct, we also provide strong baselines for benchmarking uncertainty estimation in imbalanced regression.
>
> From a technical perspective, VIR is substantially different from LDS + FDS + DER. Specifically,
>
> **(1)** VIR is a deep generative model to define how imbalanced data are generated, which is learned by a principled variational inference algorithm. In contrast, both LDS + FDS and DER are simply discriminative models (without any principled generative model formulation) that directly predict the labels from input. They are therefore easier to overfit the data.
>
> **(2)** LDS + FDS uses deterministic representations, with one vector as the final representation for each data point. In contrast, our VIR uses probabilistic representations, with one vector as the *mean* of the representation and another vector as the *variance* of the representation. Such dual representation is more robust to noise and therefore leads to better prediction performance.
>
> **(3)** LDS + FDS is a deterministic model, while our VIR is a Bayesian model. Essentially VIR is equivalent to sampling infinitely many predictions for each input data point and averaging these predictions. Therefore intuitively it makes sense that VIR could lead to better prediction performance.
>
> **(4)** DER is an uncertainty estimation model without any consideration on the imbalanced setting, which is VIR's focus. DER assigns weights according to the population and therefore tends to over-emphasize majority data and ignore minority data. In contrast, VIR introduces a reweighting mechanism naturally through the pseudo-count formulation in the NIG distribution (discussed in **Q7** as well as the paragraphs **Intuition of Pseudo-Counts for VIR** and **From Pseudo-Counts to Balanced Predictive Distribution** in the paper). Note that such a reweighting is more natural and powerful than LDS + FDS since it is rooted in the probabilistic formulation.
>
> **(5)** Different from both LDS + FDS and DER, VIR's variational autoencoder formulation naturally requires it to reconstruct the input data $x$ during training, since VIR is a generative model. Such a reconstruction loss helps further prevent overfitting and therefore improves model performance in terms of both prediction and uncertainty estimation.
>
> **(6)** Empirical results comparing LDS + FDS + DER and our VIR in Table 1-4 also verify that our VIR significantly outperforms LDS + FDS + DER.
>
> **(7)** As a byproduct, we also provide strong baselines for benchmarking uncertainty estimation in imbalanced regression.
>
> In summary, we believe that our work has both sufficient novelty and a potentially significant impact.

---

> ### Author Response · Authors · 2022-11-16
> **Thank you and response to your comments [2/4]**
>
> **Q2: "Missing related work [3, 4]. Both were published before the submission deadline and the authors should discuss the difference and compare VIR with them."**
>
> Thanks for pointing us to these interesting related works. Following your suggestion, we have included related discussion in the Related Work section of the revised paper. Due to the time limit, we do not have time to run three different random seeds on their methods (just like what we did for DIR [1]), therefore:  1. For RankSim [4], we directly report one of their results in our revised Table 1 in the paper, and we can see that RankSim performs slightly better than VIR in terms of overall and many-shot metric, but our VIR's is significantly better than RankSim in terms of median and few shot metrics. For BalancedMSE [3], since the authors do not provide the results for all metrics for direct comparison, we only included discussion in Related Work section; we are currently running BalancedMSE for different metrics and will update the response if we could make it before the discussion period ends.
>
> Table A.1: New Results on Ranksim in Terms of GM (Lower is Better), Corresponding to Table 1 in the Main Paper
>
> | Comparison | Overall | Many | Median | Few |
> | :-------: | :-----: | :-------: | :---------: | :-------: |
> |FDS + Ranksim| 4.53 | 4.13 | 5.37 | 6.89 |
> |VIR (Ours)| 4.58 | 4.27 | 5.09 | 6.31 |
> |||||
>
> **Q3: "The experimental setup can be improved. First, the authors only choose two age estimation datasets whose distributions are similar. Second, the reported numbers are not very convincing (please see the next section for quality and reproducibility)."**
>
> Note that as we mentioned in Section 4, DIR [1] uses five datasets, but only four of them are public. We therefore choose to use the smallest and the largest datasets among them for comparison. Below we included new results on a third dataset STS-B-DIR (which is an NLP dataset) to further verify the superiority of our VIR.
>
> Table B.1: New Results on a Third Dataset, STS-B-DIR in Terms of MAE
>
> | Comparison | overall | many | median | few |
> | :-------: | :-----: | :-------: | :---------: | :-------: |
> |Inv| 0.825 | 0.783 | 1.004 | 0.850 |
> |DIR+DER| 0.812 | 0.757 | 1.046 | 0.842 |
> |VIR (Ours)| 0.760 | 0.718 | 0.960 | 0.732 |
> |||||
>
> Table B.2: New Results on a Third Dataset, STS-B-DIR in Terms of Spearman's Correlation
>
> | Comparison | overall | many | median | few |
> | :-------: | :-----: | :-------: | :---------: | :-------: |
> |Inv| 0.723 | 0.678 | 0.530 | 0.685 |
> |DIR+DER| 0.730 | 0.680 | 0.526 | 0.699 |
> |VIR (Ours)| 0.770 | 0.713 | 0.534 | 0.770 |
> |||||
>
> **Q4: "I’m not fully convinced by the choice of Gaussian distribution as the prior distribution. For some specific datasets with regression labels like ages, this may possibly be a suitable assumption. But the authors claim the contributions to the task of general deep imbalanced regression, so I encourage the authors to conduct experiments on other types of regression datasets. It will strongly improve the contribution of this work if the authors can consider more datasets proposed in [1] or [3]."**
>
> This is a good question. We believe our inclusion of a third dataset STS-B-DIR (which is an NLP dataset) could further verify the superiority of our VIR. We have also included the results in the revised Appendix.
>
> **Q5: "the overall method is more like a combination of FDS [1] and DER [2] to me in the current presentation. It could be better motivated why probabilistic latent code is better than FDS to improve deep imbalanced regression."**
>
> This is a good question and highly related to **Q1** above. Please kindly refer to the response to **Q1** for detailed response.

---

> ### Author Response · Authors · 2022-11-16
> **Thank you and response to your comments [3/4]**
>
> **Q6: "The SOTA results are much worse than the ones reported in the original paper [1]. For example, in Table 1, LDS+FDS: MAE all from original 7.55 to reported 7.82, GM all from original 4.72 to reported 5.01. Also in Table 2, LDS+FDS: MAE all from 7.78 to 8.08, GM all from 4.37 to 4.66. Did the authors reproduce all results? If the SOTA is not robust and the variance between different runs is high, please consider reporting the error bar for both methods (SOTA and VIR). It is not a good practice to replace the prior work results with single runs."**
>
> We would like clarify that we use the exact same code provided by the authors of [1],  with exactly the same hyperparameter configurations, to reproduce the results of LDS+FDS.
>
> Results of deep learning models tend to be sensitive to software versions (e.g., PyTorch versions), CUDA versions, GPU types (e.g., RTX 3090 versus Tesla A100), and random seeds; this is also the case for LDS+FDS [1], and is observed by [3] as well. In our preliminary experiments, we run LDS+FDS [1] using the exact same code provided by the authors on three different types of GPUs and see different results. **The difference in terms of accuracy due to different GPU types is roughly in the range of $0\sim 0.5$ in terms of metrics such as MAE and GM.**
>
> Therefore, for fair comparison (i.e., to ensure that the performance of all the experiments will be different only because of the model design, not the hardware/software versions) we re-run each model in DIR [1] for three different random seeds on our GPU machines and software environment, and report the average performance in our paper. *The different between our re-run results and the results in the original paper [1] is below $0.3$, which is within the range of variance caused by different hardware/software versions (as tested in our preliminary experiments).* We therefore believe that our results are valid.
>
> According to your suggestion, below we provided additional results after running SQINV and LDS+FDS for 5 different random seeds (see the revised Appendix for more added results) using exactly the same code from DIR authors [1] with the same configuration. We can see that once we run all experiments in the same machine with the same software environments, the standard deviation (error bar) is below $0.05$, substantially lower than the standard deviation of running models on different machines / GPU types. This also verifies that our re-run results are valid.
>
> We have added the discussion above in the revised paper as suggested.
>
> Table C.1: Results (MAE) of Running SQINV for 5 Different Random Seeds on AgeDB
>
> | Comparison | Overall | Many | Median | Few |
> | :-------: | :-----: | :-------: | :---------: | :-------: |
> |SQINV 1| 8.04 | 7.40 | 9.01 | 11.33 |
> |SQINV 2| 8.12 | 7.47 | 9.17 | 11.58 |
> |SQINV 3| 8.21 | 7.59 | 9.01 | 11.81 |
> |SQINV 4| 7.94 | 7.39 | 8.58 | 11.39 |
> |SQINV 5| 7.99 | 7.47 | 8.98 | 11.49 |
> |SQINV avg| 8.06 | 7.46 | 8.95 | 11.52 |
> |SQINV std| 0.01 | 0.01 | 0.04 | 0.03 |
> |||||
>
>
> Table C.2: Results (MAE) of Running LDS+FDS for 5 Different Random Seeds on AgeDB
>
> | Comparison | Overall | Many | Median | Few |
> | :-------: | :-----: | :-------: | :---------: | :-------: |
> |LDS+FDS 1| 7.87 | 7.26 | 8.97 | 10.88 |
> |LDS+FDS 2| 7.98 | 7.44 | 8.77 | 11.16 |
> |LDS+FDS 3| 8.07 | 7.54 | 8.71 | 11.41 |
> |LDS+FDS 4| 8.02 | 7.50 | 8.82 | 11.34 |
> |LDS+FDS 5| 7.93 | 7.35 | 8.80 | 10.96 |
> |LDS+FDS avg| 7.97 | 7.42 | 8.81 | 11.15 |
> |LDS+FDS std| 0.01 | 0.01 | 0.01 | 0.04 |
> |||||

---

> ### Author Response · Authors · 2022-11-16
> **Thank you and response to your comments [4/4]**
>
> **Q7: "[-] I’m concerned about the conditional log-likelihood $L^{D}$ in Eq. (1). Did the authors have any ablation study on the weighting parameter $\lambda$? "**
>
> Thanks for the suggestion. Following your suggestion, we added an ablation study on $\lambda=10.0, 1.0, 0.1, 0.01, 0.001$ below (more results are in our revised Appendix marked in blue) to show the difference, and we can also conclude that when $\lambda=0.1$, our model achieves the best performance. Therefore, we set the value of $\lambda$ to 0.1.
>
> Table D.1: New Ablation Study on $\lambda$ in Terms of NLL
>
> | Comparison | overall | many | median | few |
> | :-------: | :-----: | :-------: | :---------: | :-------: |
> |$\lambda=10.0$| 3.827 | 3.733 | 4.140 | 4.407 |
> |$\lambda=1.0$| 3.848 | 3.738 | 4.041 | 4.356 |
> |$\lambda=0.1$| 3.785 | 3.694 | 3.963 | 4.151 |
> |$\lambda=0.01$| 3.887 | 3.797 | 4.007 | 4.401 |
> |$\lambda=0.001$| 3.980 | 3.868 | 4.161 | 4.546 |
> |||||
>
> Table D.2: New Ablation Study on $\lambda$ in Terms of MSE
>
> | Comparison | overall | many | median | few |
> | :-------: | :-----: | :-------: | :---------: | :-------: |
> |$\lambda=10.0$| 104.31 | 91.01 | 116.43 | 196.35 |
> |$\lambda=1.0$| 104.10 | 87.28 | 128.26 | 196.12 |
> |$\lambda=0.1$| 86.28 | 76.87 | 101.57 | 132.90 |
> |$\lambda=0.01$| 86.86 | 76.58 | 99.95 | 147.82 |
> |$\lambda=0.001$| 87.25 | 74.13 | 104.78 | 162.64 |
> |||||
>
>
> **Q8: "The overall task is discriminative and evaluated on the accuracy of predicting the regression label. The weighting of this conditional log-likelihood term should be discussed. This term can dominate the objective since the authors use image-based age estimation datasets with high-dimensional x."**
>
> For fair comparison, we use the best reweighting method in DIR [1] and use SQINV (for AgeDB and IMDB-WIKI) and INV (for STS-B) for all evaluated methods. We have included the clarification above in the revision as suggested.
>
>
> [1] Yang et al., Delving into Deep Imbalanced Regression. ICML 2021
>
> [2] Amini et al., Deep Evidential Regression. NeurIPS 2020
>
> [3] Ren et al., Balanced MSE for Imbalanced Visual Regression, CVPR 2022
>
> [4] Gong et al., RankSim: Ranking Similarity Regularization for Deep Imbalanced Regression, ICML 2022

---

> > ### Comment · Reviewer_5JWG · 2022-11-18
> > **Response to the authors**
> >
> > Thanks for the effort to address my concerns. However, the experimental parts are still not sufficient, so I keep my rating. I encourage the authors to extensively improve the experiments to be accepted in a top-tier venue. Also, the authors can consider attaching your code in the reviewing process.
> >
> > I have a few more comments below,
> >
> > (1) In the Table C.1 and C.2 from the authors' response, the standard deviation is wrongly calculated -- e.g. in Table C.1 overall std should be 0.095 (not 0.01).
> > I also run the same code of the baseline on my own computing resources but did not observe such a performance drop. It may have some cases where different machines produce different results with the same code, so I encourage the authors to make more efforts to add std to each number (especially for your proposed method VIR, not only baselines as you replied) and update the manuscript accordingly. The current status of the manuscript is still not empirically convincing to me.
> >
> > (2) I believe other benchmarks besides the age-estimation dataset are not only bonus points but are very necessary to this task. AgeDB-DIR and IMDB-WIKI-DIR share nearly the same imbalanced pattern, which makes VIR an overclaim for imbalanced regression. So including a complete and fair comparison between VIR and baselines on other datasets is important. I appreciate the authors' effort to run the preliminary results of STS-B-DIR. But an extensive and complete version (with error bars) of Table B.1 and B.2 in the updated manuscript is needed (like Table 1 and 2 in the original submission). This is also why I do not think the current manuscript is ready to be published and actionable during the reviewing process.

---

### Official Review · Reviewer_ddev · 2022-10-25

**Confidence:** 3
**Correctness:** 2
**Technical Novelty And Significance:** 3
**Empirical Novelty And Significance:** 3
**Recommendation:** 6

**Clarity, Quality, Novelty And Reproducibility:**

*Clarity
- Are there any drawbacks of computational cost since the proposed method requires to use of multiple x even when testing?
- Why IID assumption harms performance for data with minority labels?
- There is no definition for Omega in Eq. 6.
- Section 3.4 is hard to follow. For example, there is no definition for the parameters of NIG at the point of Eq. 6.
- What is the difference between LDS + FDS + DER  and the proposed method?
- Tables 5 and 6 are confusing. There is no result of "eoncoder only" in Table 6. Why?

*Quality
- Please see the above comments.

*Novelty
- The proposed method seems to be novel.

*Reproducibility
- Code is not available, and the description of the proposed method is also incomplete.


**Strength And Weaknesses:**

*Strength
- The authors simultaneously address the label imbalance problem and uncertainty qualification capability in regression, which is a novel problem setting.
- The non-trivial combination of two different approaches is proposed and works well.

*Weaknesses
- The intuition that each of the components of the proposed method works for their purpose is not described well in the main text. For example, there is no overall procedure for their reweighting technique, which is provided in the paper (Yang et al., 2021) though.
- In Section 3.3, the authors proposed to use the covariance of z for probabilistic overall covariance. However, it is unclear why they do not use the covariance of z^{mu} and z^{sigma} instead.
- There is no intuition why the proposed approach outperformed DIR, which could be more practical in terms of prediction performance in some cases, in my understanding.

**Summary Of The Paper:**

This paper simultaneously addresses the label imbalance problem and uncertainty qualification capability in regression.
The authors propose to enhance the reweighting technique dealing with the imbalance problem in (Yang et al., 2021) to be applicable to VAE and combine the output distribution and the corresponding loss in (Amini et al., 2020), which provides uncertainty qualification capability.
Experimental results on several real-world datasets demonstrate that the proposed method performs better than state-of-the-art imbalanced regression models in terms of both accuracy and uncertainty estimation.

**Summary Of The Review:**

Although the proposed method is a good combination of the two SOTA methods and performs well, the description of the proposed method does not seem self-complete.

---

> ### Author Response · Authors · 2022-11-16
> **Thank you and response to your comments [1/3]**
>
> Thank you for your valuable comments. We are glad that you find our problem ``"novel"`` and our method ``"non-trivial and works well"``. Below we address your questions one by one.
>
> **Q1: "For example, there is no overall procedure for their reweighting technique, which is provided in the paper (Yang et al., 2021) though."**
>
> Thanks for mentioning this and we are sorry for the confusion. As we discussed in Sec. 4, we used the reweighting scheme SQINV (i.e., square root inverse) for all the experiments. According to your suggestion, we have added more details for SQINV in our revision.
>
> **Q2: "In Section 3.3, the authors proposed to use the covariance of z for probabilistic overall covariance. However, it is unclear why they do not use the covariance of z^{mu} and z^{sigma} instead."**
>
> This is a good question. Suppose $z$ is deterministic, then the overall covariance represents how scattered all $z$'s in each bins are; this is why DIR [1] uses the covariance directly. However, in our work, $z$ is probabilistic, therefore we need to calculate the probabilistic overall covariance. Specifically, we need to compute the *mean of the overall covariance* and *the variance of the overall covariance* (see Section 3.3 for more details).
>
> Note that using the covariance of $z^{mu}$ and $z^{sigma}$ does not work, because $z^{mu}$ and $z^{sigma}$ represent the *uncertainty* of *a single data point*, not the ``scatteredness'' of *different $z$'s* in the same bins.
>
> **Q3: "There is no intuition why the proposed approach outperformed DIR, which could be more practical in terms of prediction performance in some cases ..."**
>
> **(1)** DIR uses deterministic representations, with one vector as the final representation for each data point. In contrast, our VIR uses probabilistic representations, with one vector as the *mean* of the representation and another vector as the *variance* of the representation. Such dual representation is more robust to noise and therefore leads to better prediction performance.
>
> **(2)** DIR is a deterministic model, while our VIR is a Bayesian model. Essentially VIR is equivalent to sampling infinitely many predictions for each input data point and averaging these predictions. Therefore intuitively it makes sense that VIR could lead to better prediction performance.
>
> **(3)** It is also worth noting that DIR is a deterministic model and therefore cannot produce uncertainty estimation. In contrast, Our VIR formulates a probabilistic deep generative model for imbalanced data, and therefore can naturally produce both more accurate predictions compared to DIR [1] and better uncertainty estimation compared to DER [2].
>
>
> **Q4: "Are there any drawbacks of computational cost since the proposed method requires to use of multiple x even when testing?"**
>
> We are sorry for the confusion. Actually our VIR does *not* need to use multiple $x$'s when testing. The computation of statistics (mean and variance) from multiple $x$'s is only needed during training. During testing, our VIR directly uses these statistics collected during training and therefore does not need to re-compute them. We have included the clarification above in the Sec. 3.3 of the revised paper as suggested.

---

> ### Author Response · Authors · 2022-11-16
> **Thank you and response to your comments [2/3]**
>
> **Q5: "Why IID assumption harms performance for data with minority labels?"**
>
> This is a good question. Intuitively the IID assumption treats data with different labels separately; in contrast, our NID assumption allows the model to borrow from data with similar labels (e.g., label 20 and label 21 in terms of age estimation) both during training and testing when making predictions for minority data. This turns out to be crucial for improving performance for data with minority labels. Beyond the intuition above, we also provide some theoretical insight on why NID is better than IID in the imbalanced setting from the perspective of generalization error below.
>
> **Generalization Error, Bias, and Variance.** We could analyze the generalization error of our VIR by bounding the generalization with the sum of three terms: (a) the bias of our estimator, (2) the variance of our estimator, (3) model complexity. Essentially VIR uses the N.I.D. assumption to increase our estimator's bias, but significantly reduces its variance in the imbalanced setting. Since the model complexity is kept the same (using the same backbone neural network) as the baselines, N.I.D. will lead to a lower generalization error.
>
> **Variance of Estimators in Imbalanced Settings.** In the imbalanced setting, one typically use inverse weighting to produce an unbiased estimator (i.e., making the first term of the aforementioned bound zero). However, for data with extremely low density, its inverse would be extremely large, therefore leading to a very large variance for the estimator. Our VIR replaces I.I.D. with N.I.D. to ``smooth out'' such singularity, and therefore significantly lowers the variance of the estimator (i.e., making the second term of the aforementioned bound smaller), and ultimately lowers the generalization error.
>
> **Q6: "There is no definition for Omega in Eq. 6."**
>
> Thanks for mentioning this. We have included the definition of $\Omega$ immediately below Eq. 6 in the revised paper.
>
> **Q7: "Section 3.4 is hard to follow. For example, there is no definition for the parameters of NIG at the point of Eq. 6."**
>
> The definition of parameters in NIG follows the classic PRML book (Bishop, 2006). At a high level, the normal inverse gamma (NIG) distribution defines how a Gaussian distribution's parameters $\mu$ and $\Sigma$ are generated. Specifically, (1) $\Sigma$ is first generated from an *inverse gamma distribution* $\Sigma \sim \Gamma^{-1}(\alpha,\beta)$, where $\alpha$ and $\beta$ are the inverse gamma distribution's *two* parameters. (2) Then, $\mu$ is generated from a *normal distribution* $\mu \sim N(\gamma,\Sigma/\nu)$, where $\Sigma$ is the previously generated covariance, and $\gamma$ and $\nu$ are *two* additional parameters for the normal distribution. Combining the *inverse gamma distribution* and the *normal distribution* above, we then have the *normal inverse gamma* distribution, denoted as $NIG(\gamma,\nu,\alpha,\beta)$, with *four* parameters $\gamma$, $\nu$, $\alpha$, and $\beta$.

---

> ### Author Response · Authors · 2022-11-16
> **Thank you and response to your comments [3/3]**
>
> **Q8: "What is the difference between LDS + FDS + DER and the proposed method?"**
>
> This is a good question. VIR is substantially different from LDS + FDS + DER. Specifically,
>
> **(1)** VIR is a deep generative model to define how imbalanced data are generated, which is learned by a principled variational inference algorithm. In contrast, both LDS + FDS and DER are simply discriminative models (without any principled generative model formulation) that directly predict the labels from input. They are therefore easier to overfit the data.
>
> **(2)** LDS + FDS uses deterministic representations, with one vector as the final representation for each data point. In contrast, our VIR uses probabilistic representations, with one vector as the *mean* of the representation and another vector as the *variance* of the representation. Such dual representation is more robust to noise and therefore leads to better prediction performance.
>
> **(3)** LDS + FDS is a deterministic model, while our VIR is a Bayesian model. Essentially VIR is equivalent to sampling infinitely many predictions for each input data point and averaging these predictions. Therefore intuitively it makes sense that VIR could lead to better prediction performance.
>
> **(4)** DER is an uncertainty estimation model without any consideration on the imbalanced setting, which is VIR's focus. DER assigns weights according to the population and therefore tends to over-emphasize majority data and ignore minority data. In contrast, VIR introduces a reweighting mechanism naturally through the pseudo-count formulation in the NIG distribution (discussed in **Q7** as well as the paragraphs **Intuition of Pseudo-Counts for VIR** and **From Pseudo-Counts to Balanced Predictive Distribution** in the paper). Note that such a reweighting is more natural and powerful than LDS + FDS since it is rooted in the probabilistic formulation.
>
> **(5)** Different from both LDS + FDS and DER, VIR's variational autoencoder formulation naturally requires it to reconstruct the input data $x$ during training, since VIR is a generative model. Such a reconstruction loss helps further prevent overfitting and therefore improves model performance in terms of both prediction and uncertainty estimation.
>
> **(6)** Empirical results comparing LDS + FDS + DER and our VIR in Table 1-4 also verify that our VIR significantly outperforms LDS + FDS + DER.
>
> **Q9: "Tables 5 and 6 are confusing. There is no result of "encoder only" in Table 6. Why?"**
>
> We are sorry for the confusion. Note that Table 5 is an ablation study for *prediction accuracy*, while Table 6 is an ablation study for *uncertainty estimation*. A model consists of an encoder and a predictor. The "encoder only" ablation model contains only the encoder part of our VIR discussed in Section 3.3 and uses a baseline predictor; this ablation model's predictor therefore only generates a deterministic value as the final prediction and cannot perform uncertainty estimation; this is why it is impossible to include "encoder only" in Table 6 (which is for *uncertainty estimation*). Note that in the revised version, Table 5 and 6 are moved to the Appendix due to space constraint after incorporating all reviewers' suggested changes.
>
> **Q10: "Code is not available, and the description of the proposed method is also incomplete."**
>
> We have cleaned up the code base and will release it upon the acceptance of the paper.
>
>
> [1] Yang et al., Delving into Deep Imbalanced Regression. ICML 2021
>
> [2] Amini et al., Deep Evidential Regression. NeurIPS 2020

---

### Official Review · Reviewer_bYud · 2022-10-28

**Confidence:** 2
**Correctness:** 3
**Technical Novelty And Significance:** 2
**Empirical Novelty And Significance:** 2
**Recommendation:** 6

**Clarity, Quality, Novelty And Reproducibility:**

This paper is well written. The method is well-motivated. The experimental results well support the proposed method.

**Strength And Weaknesses:**

Strength:

This method uses a probabilistic representation approach that can quantify the uncertainty in the estimated prediction model. In addition, the method can potentially work well for the minority group in the data.


Weakness:

I think the paper can benefit from providing more practical guidance on implementing the proposed method. For example, how to select the number of bins B in Section 3.1? Why is it a good idea to partition the label space $\mathcal{Y}$ into equal-interval bins, but not equal-size bins (with the same number of data points)? How to select the bandwidth in the kernel function $k(\cdot)$ to compute the neighboring data and smoothed statistics in Section 3.3? How does the bandwidth vary with the sample size N?

In addition, can the authors provide some theoretical guarantee (e.g., unbiasedness, consistency) on the proposed method?

**Summary Of The Paper:**

This paper proposes a probabilistic deep learning model, namely variational imbalanced regression (VIR), when the distribution of outcome is imbalanced. This method uses data with similar outcomes to estimate the distribution of the latent representation, provides a point estimate of the outcome, and provides uncertainty quantification of the point estimate. The method can work well for the minority group in the data.

**Summary Of The Review:**

This paper provides a solution to address the problem of probabilistic deep imbalanced regression. The proposed method can potentially accurately estimate the outcome and quantify the uncertainty in the estimation. I think the paper can benefit from providing more practical guidance on using the proposed method, and providing some theoretical guarantee on the proposed method.

---

> ### Author Response · Authors · 2022-11-16
> **Thank you and response to your comments**
>
> Thank you for your valuable comments. We are glad that you find our method ``"well-motivated"`` and our results ``" well support"``. Below we address your questions one by one.
>
> **Q1: "how to select the number of bins B in Section 3.1?"**
>
> For fair comparison, we directly follow DIR when constructing bins. For example, in the AgeDB dataset, since the range of the labels (people's ages) is between 1 and 100, it is natural to use 100 bins, each representing one year of age. We have also revised Sec. 3.1 accordingly as suggested.
>
> **Q2: "Why is it a good idea to partition the label space $\mathcal{Y}$ into equal-interval bins, but not equal-size bins (with the same number of data points)?"**
>
> This is a good question. Note that since our smoothing kernel function is based on labels (i.e., $k(y, y')$); therefore it is more reasonable to use equal-interval bins rather than equal-size bins.
>
> **(1)** For example, if we use the equal-interval bins $[0,1),[1,2),...$, VIR will naturally compute $k(y, y')$ for $y=1,2,3,4,5,...$ and $y'=1,2,3,4,5,...$.
>
> **(2)** In contrast, if we use equal-size bins, VIR may end up with *large intervals* and may lead to inaccurate kernel values for $k(y, y')$. To see this, consider a case where equal-size bins are $[0,1),[1,2),[2,3.1),[3.1,8.9),...$; the kernel value $k(y, y')$ between bins $[2,3.1)$ and $[3.1,8.9)$ is $k(2,3.1)$, which is very inaccurate since $3.1$ is very far away from the mean of the bin $[3.1,8.9)$ (i.e., $6$). Using small and equal-interval bins can naturally address such issues.
>
> **Q3: "How to select the bandwidth in the kernel function $k(\cdot)$ to compute the neighboring data and smoothed statistics in Section 3.3?"**
>
> For fair comparison, we use the same bandwidth as the DIR paper.
>
> **Q4: "How does the bandwidth vary with the sample size N?"**
>
> Our bandwidth is fixed for different sample size N.
>
> **Q5: "In addition, can the authors provide some theoretical guarantee (e.g., unbiasedness, consistency) on the proposed method?"**
>
> This is a good point and would also be interesting future work. We do have some initial thoughts in terms of the theoretical guarantees and would love to discuss with the reviewer during the discussion period to get more feedback.
>
> **Generalization Error, Bias, and Variance.** We could analyze the generalization error of our VIR by bounding the generalization with the sum of three terms: (a) the bias of our estimator, (2) the variance of our estimator, (3) model complexity. Essentially VIR uses the N.I.D. assumption to increase our estimator's bias, but significantly reduces its variance in the imbalanced setting. Since the model complexity is kept the same (using the same backbone neural network) as the baselines, N.I.D. will lead to a lower generalization error.
>
> **Variance of Estimators in Imbalanced Settings.** In the imbalanced setting, one typically uses inverse weighting to produce an unbiased estimator (i.e., making the first term of the aforementioned bound zero). However, for data with extremely low density, its inverse would be extremely large, therefore leading to a very large variance for the estimator. Our VIR replaces I.I.D. with N.I.D. to ''smooth out'' such singularity, and therefore significantly lowers the variance of the estimator (i.e., making the second term of the aforementioned bound smaller), and ultimately lowers the generalization error.

---

### Official Review · Reviewer_EUDh · 2022-10-31

**Confidence:** 3
**Correctness:** 3
**Technical Novelty And Significance:** 3
**Empirical Novelty And Significance:** 3
**Recommendation:** 6

**Clarity, Quality, Novelty And Reproducibility:**

The paper is somewhat clear, but some details are missing. This paper is technically reasonable.

**Strength And Weaknesses:**

Strength

- The problem studied in this paper is interesting and valuable. Both balanced accuracy and uncertainty estimation are important topics in regression.
- The paper is well-organized and clearly written, which is easy to follow.
- This paper provides some novel perspectives for probabilistic deep imbalanced regression. Leveraging Neighboring and Identically Distributed data is a valuable attempt for handling imbalance and uncertainty issues.

Weaknesses

- In Section 3, the authors declare some limitations of LDS and DER. However, the descriptions are somewhat confusing. In my opinion, the authors should provide theoretical perspectives or specific cases for the limitations, thus illustrating the advantage of VIR.
- Since the I.I.D. assumption has been replaced with the N.I.D. assumption, some theoretical guarantees could be provided to further demonstrate the effectiveness of the algorithm.
- The impact of $\lambda$ is unclear. Further ablation studies should be conducted to illustrate the impact of $\lambda$.
- One can observe in Table 4 that the uncertainty estimation performance of the proposed VIR is worse than LDS+FDS+DER on Medium labels. The authors should further analyze the reasons for this phenomenon.


**Summary Of The Paper:**

This paper proposes a novel approach named variational imbalanced regression for the probabilistic deep imbalanced regression problem. It leverages data with similar regression labels to compute the latent representation’s variational distribution, and predicts the entire normal-inverse-gamma distributions and modulates the associated conjugate distributions to impose probabilistic reweighting on the imbalanced data.

**Summary Of The Review:**

This paper provides some new perspectives for imbalanced regression. However, I have concerns about whether this work can be accepted in its current form. I will update my reviews based on the authors' responses.

---

> ### Author Response · Authors · 2022-11-16
> **Thank you and response to your comments [1/2]**
>
> Thank you for your valuable comments. We are glad that you find the problem we studied ``"interesting and valuable"``, our paper ``"well-organized and clearly written"``, and our perspectives ``"novel"``. Below we address your questions one by one.
>
> **Q1: "In Section 3, the authors declare some limitations of LDS and DER. However, the descriptions are somewhat confusing. In my opinion, the authors should provide theoretical perspectives or specific cases for the limitations, thus illustrating the advantage of VIR."**
>
> Indeed, thoroughly studying our VIR from the theoretical perspective would be interesting future work. Here we provide some preliminary theoretical insights. (1) DIR [1] is a deterministic model and therefore cannot produce uncertainty estimation, (2) While DER [2] can naturally produce uncertainty estimation, it does not take into account the imbalance in data, and therefore performs poorly in imbalanced settings, as shown in Table 1 & 2 by comparing SQINV vs. DER (i.e., SQINV + DER)]. (3) In contrast, Our VIR formulates a probabilistic deep generative model for imbalanced data, and therefore can naturally produce both more accurate predictions compared to DIR [1] and better uncertainty estimation compared to DER [2].
>
> **Q2: "Since the I.I.D. assumption has been replaced with the N.I.D. assumption, some theoretical guarantees could be provided to further demonstrate the effectiveness of the algorithm."**
>
> This is a good point and would also be interesting future work. We do have some initial thoughts in terms of the theoretical guarantees and would love to discuss with the reviewer during the discussion period to get more feedback.
>
> **Generalization Error, Bias, and Variance.** We could analyze the generalization error of our VIR by bounding the generalization with the sum of three terms: (a) the bias of our estimator, (2) the variance of our estimator, (3) model complexity. Essentially VIR uses the N.I.D. assumption to increase our estimator's bias, but significantly reduces its variance in the imbalanced setting. Since the model complexity is kept the same (using the same backbone neural network) as the baselines, N.I.D. will lead to a lower generalization error.
>
> **Variance of Estimators in Imbalanced Settings.** In the imbalanced setting, one typically uses inverse weighting to produce an unbiased estimator (i.e., making the first term of the aforementioned bound zero). However, for data with extremely low density, its inverse would be extremely large, therefore leading to a very large variance for the estimator. Our VIR replaces I.I.D. with N.I.D. to ``smooth out'' such singularity, and therefore significantly lowers the variance of the estimator (i.e., making the second term of the aforementioned bound smaller), and ultimately lowers the generalization error.
>
> We included the discussion above in our revised paper (Sec. 3.6 and Sec. A of the Appendix) as suggested.

---

> ### Author Response · Authors · 2022-11-16
> **Thank you and response to your comments [2/2]**
>
> **Q3: "The impact of $\lambda$ is unclear. Further ablation studies should be conducted to illustrate the impact of $\lambda$."**
>
> Thank you for the suggestion. For the importance of the regularization term in an experiment, we reported a small ablation study on $\lambda=10.0, 1.0, 0.1, 0.01, 0.001$ below (more results are in our revised Appendix marked in blue) to show the difference, and we can also conclude that when $\lambda=0.1$, our model achieves the best performance. Therefore, we set the value of $\lambda$ to 0.1.
>
> Table A.1: New Ablation Study on $\lambda$ in Terms of NLL
>
> | Comparison | overall | many | median | few |
> | :-------: | :-----: | :-------: | :---------: | :-------: |
> |$\lambda=10.0$| 3.827 | 3.733 | 4.140 | 4.407 |
> |$\lambda=1.0$| 3.848 | 3.738 | 4.041 | 4.356 |
> |$\lambda=0.1$| 3.785 | 3.694 | 3.963 | 4.151 |
> |$\lambda=0.01$| 3.887 | 3.797 | 4.007 | 4.401 |
> |$\lambda=0.001$| 3.980 | 3.868 | 4.161 | 4.546 |
> |||||
>
> Table A.2: New Ablation Study on $\lambda$ in Terms of MSE
>
> | Comparison | overall | many | median | few |
> | :-------: | :-----: | :-------: | :---------: | :-------: |
> |$\lambda=10.0$| 104.31 | 91.01 | 116.43 | 196.35 |
> |$\lambda=1.0$| 104.10 | 87.28 | 128.26 | 196.12 |
> |$\lambda=0.1$| 86.28 | 76.87 | 101.57 | 132.90 |
> |$\lambda=0.01$| 86.86 | 76.58 | 99.95 | 147.82 |
> |$\lambda=0.001$| 87.25 | 74.13 | 104.78 | 162.64 |
> |||||
>
> **Q4: "One can observe in Table 4 that the uncertainty estimation performance of the proposed VIR is worse than LDS+FDS+DER on Medium labels. The authors should further analyze the reasons for this phenomenon."**
>
> This is a good question. There is a natural trade-off among many-shot, medium-shot, and few-shot performance. The key hyperparameter that affects such a trade-off is the smoothing kernel that is used to compute Eq. (4) in the paper. Intuitively, different smoothing kernels will lead to different weights for reweighting data points in the many-shot, medium-shot, and few-shot areas, and therefore affect the trade-off. In our experiments, we used a default smoothing kernel used in DIR [1], but we believe that it is possible to search for the optimal smoothing kernel to achieve the optimal trade-off where VIR can outperform the baselines even in the medium-shot areas (note that in table 3 our VIR's medium result is already better than baselines even with the default kernel).
>
> Besides, it is also worth noting that our VIR focuses on imbalanced regression. Therefore we are more interested in seeing improvement on ``overall`` and ``few-shot`` performance.
>
>
> [1] Yang et al., Delving into Deep Imbalanced Regression. ICML 2021
>
> [2] Amini et al., Deep Evidential Regression. NeurIPS 2020

---

### Official Review · Reviewer_xTgV · 2022-11-01

**Confidence:** 4
**Correctness:** 2
**Technical Novelty And Significance:** 2
**Empirical Novelty And Significance:** 2
**Recommendation:** 3

**Clarity, Quality, Novelty And Reproducibility:**

The paper is quite clear even if it is sometimes hard to read with quite dense mathematical notations (e.g. eq. 2, 3, 4) and some unclear cross references (i.e. "see below" statements). The paper combines existing techniques (smoothed statistics and conjugate distribution parametrization)  to solve the important problem of prediction on imbalanced datasets. However, it misses many important related works which would allow to more correctly estimate the novelty of the paper for the reader.

**Strength And Weaknesses:**

Pros:

1. The task of uncertainty estimation for regression on imbalanced dataset is important.
2. The idea of combining smoothed statistics to fix imbalanced dataset and produce uncertainty estimates via parametrizing conjugate distributions is interesting.
3. The method achieves great improvement on 2 datasets.

##########################################################################

Cons:

- Related works: The paper misses many important related works for uncertainty estimation for regression like [1, 2, 3, 4, 5, 6, 7] but also many others. This includes very common baselines like Dropout [6] and Ensemble [1] which work for regression. The paper also does not discuss GP-based methods [2, 3] which can be used for regression. Further, the paper does not discuss Posterior Networks methods [4, 5, 7]. More specifically, [4, 5] shares many similarities with VIR. They use conjugate distributions, pseudo-count interpretations, posterior updates, and variational losses which are key parts of VIR. NatPN is also designed to work for regression. Action suggestion: discuss all of these methods in the related work section.
- Desiderata: The two desiderata are sometimes not clear. It feels that the two desiderata are high quality *uncertainty estimation* and high performance on *imbalanced dataset*. However the paper mentions that the desiderata are “performance improvement and uncertainty estimation”. The paper also phrases the identification of the desiderata of imbalanced dataset and uncertainty estimation as a contribution.  However people already looked at imbalanced datasets and uncertainty estimation separately in previous works. Finally, the paper does not discuss existing desiderata for uncertainty estimation [8, 7, 9]. Action suggestion: I would recommend to change the phrasing of this contribution and have a discussion on existing desiderata in uncertainty estimation.
- Loss: I felt that the description of the loss was sometimes confusing. I would be interested in the derivation fo the loss. This could also be provided in the appendix. In eq.(1), what is p_\theta(z_i) ? Does this ELBO loss assume any prior ? What is the importance of the regularization loss in the total loss ? Action suggestion: Beyond the answer to these questions, it would be intresting to show the important of the regularization term in an experiment.
- Clarity: The paper is sometimes hard to read. E.g. the paper mentions multiple times “see below” which leaves unclear where to exactly find the missing information. Action suggestion: make explicit reference when pointing to further details.
- Bins: It is unclear what is the sensitivity of the method to the number of bins. It was also unclear to me when readin sec. 3.1. what data are used to build the mean representation from the bins. Action suggestion: I would recommend to clarify the bins construction since it is an essential part of the method. I would also be interested in an experiment on the number of bins.
- Experiences: The experiments do not look very extensive to me. They consider only two datasets and a single backbone network. They do not compare to DropOut, Ensemble, GP, or NatPN which would be appropriate baselines. They do not look at common uncertainty estimation metrics like OOD detection scores. They also do not provide error bars which are key to assess the significance of the results. Action suggestion: I would recommend to add at least one dataset (e.g. depth estimation) with another backbone architecture. I would also recommend to add at least Ensemble which is a common and powerful baseline and NatPN which shares many similarities with VIR. I would also recommend to add error bars.

I am happy to improve my score if a majority of the above points are addresses (e.g. with the action suggestions).

[1] Simple and scalable predictive uncertainty estimation using deep ensembles, NeurIPS 2017

[2] On feature collapse and deep kernel learning for single forward pass uncertainty.

[3] Simple and principled uncertainty estimation with deterministic deep learning via distanceawareness, NeurIPS 2020.

[4] Posterior Network: uncertainty estimation without ood samples via density-based pseudo-counts. NeurIPS 2020

[5] Natural Posterior Network: deep bayesian uncertainty for exponential family distributions, ICLR 2022

[6] Dropout as a bayesian approximation: representing model uncertainty in deep learning, ICML 2016

[7] Graph Posterior Network: bayesian predictive uncertainty for node classification. NeurIPS 2021.

[8] Can you trust your model’s uncertainty? evaluating predictive uncertainty under dataset shift, NeurIPS 2019.

[9] NOMU: Neural Optimization-based Model Uncertainty. ICML 2022

**Summary Of The Paper:**

The paper looks at uncertainty estimation for regression on imbalanced datasets. It proposes a new method called VIR which combines variational inference, smoothed statistics, and conjugate distribution parametrization. In the experiments, VIR is evaluated on 2 imbalanced datasets on accuracy and calibration metrics.

**Summary Of The Review:**

Overall, I vote for strong reject. The task is important and well motivated and the . My major concerns are about the related work, and the experiences (see cons beow). Hopefully the authors can address my concern in the rebuttal period.

---

> ### Author Response · Authors · 2022-11-16
> **Thank you and response to your comments [1/2]**
>
> Thank you for your valuable comments. We are glad that you find the problem we identified ``"important"``, our idea ``"interesting"``, and our method ``"achieves great improvement on 2 datasets"``. Below we address your questions one by one.
>
> **Q1: "Related works: The paper misses many important related works for uncertainty estimation for regression like [1, 2, 3, 4, 5, 6, 7] but also many others. This includes very common baselines like Dropout [6] and Ensemble [1] which work for regression. The paper also does not discuss GP-based methods [2, 3] which can be used for regression. Further, the paper does not discuss Posterior Networks methods [4, 5, 7]. More specifically, [4, 5] shares many similarities with VIR. They use conjugate distributions, pseudo-count interpretations, posterior updates, and variational losses which are key parts of VIR. NatPN is also designed to work for regression. Action suggestion: discuss all of these methods in the related work section."**
>
> Thank you very much for pointing us to the interesting related works. We have added all the papers mentioned above into the Related Work section, especially "Posterior Networks methods [4, 5, 7]", and discussed the differences between our VIR and the added related works in the revised paper.
>
> **Q2: "Desiderata: The two desiderata are sometimes not clear. It feels that the two desiderata are high quality uncertainty estimation and high performance on imbalanced dataset. However the paper mentions that the desiderata are “performance improvement and uncertainty estimation”. The paper also phrases the identification of the desiderata of imbalanced dataset and uncertainty estimation as a contribution. However people already looked at imbalanced datasets and uncertainty estimation separately in previous works. Finally, the paper does not discuss existing desiderata for uncertainty estimation [8, 7, 9]. Action suggestion: I would recommend to change the phrasing of this contribution and have a discussion on existing desiderata in uncertainty estimation."**
>
> This is a good suggestion. According to your suggestion, we changed the phrasing of our contribution and include the discussion on existing desiderata in uncertainty estimation (in the Related Work section).
>
> **Q3: "Loss: I felt that the description of the loss was sometimes confusing. I would be interested in the derivation fo the loss. This could also be provided in the appendix. In eq.(1), what is p_\theta(z_i) ? Does this ELBO loss assume any prior ? What is the importance of the regularization loss in the total loss ? Action suggestion: Beyond the answer to these questions, it would be intresting to show the important of the regularization term in an experiment.."**
>
> We are sorry for the confusion. $p_\theta(z_i)$ in eq.(1) is the standard Gaussian prior ($N(0,I)$), which is the same as the typical VAE. We have added this part in the paper. For the importance of the regularization term in an experiment, we reported a small ablation study on $\lambda=0.1, 0.01, 0.001$ below (more results are in the Appendix) to show the difference, and we can also conclude that when $\lambda=0.1$, our model achieves the best performance. Therefore, we set the value of $\lambda$ to 0.1.
>
> Table A.1: New Ablation Study on $\lambda$ in Terms of NLL
>
> | Comparison | overall | many | median | few |
> | :-------: | :-----: | :-------: | :---------: | :-------: |
> |$\lambda=0.1$| 3.785 | 3.694 | 3.963 | 4.151 |
> |$\lambda=0.01$| 3.887 | 3.797 | 4.007 | 4.401 |
> |$\lambda=0.001$| 3.980 | 3.868 | 4.161 | 4.546 |
> |||||
>
> Table A.2: New Ablation Study on $\lambda$ in Terms of MSE
>
> | Comparison | overall | many | median | few |
> | :-------: | :-----: | :-------: | :---------: | :-------: |
> |$\lambda=0.1$| 86.28 | 76.87 | 101.57 | 132.90 |
> |$\lambda=0.01$| 86.86 | 76.58 | 99.95 | 147.82 |
> |$\lambda=0.001$| 87.25 | 74.13 | 104.78 | 162.64 |
> |||||

---

> > ### Comment · Reviewer_xTgV · 2022-11-16
> > **Thank you for the answer**
> >
> > Thank you for the answer. I really appreciate the efforts which improve the paper. However, I still feel fill that some important points need to be addressed.
> >
> > - Q5: I feel that the bin setting is still quite confusing for the regression setting. Regression is usually defined as predicting continuous labels but the experiments consider discrete bins. In particular, discrete bins are apparently more natural to consider for the AgeDB dataset. It would be interesting to consider datasets with unambiguously scalar labels since the paper focuses on regression. It is also still unclear if the definition of the number of bins is important on such datasets. Action suggestion: Take datasets with scalar labels and experiment for different number of bins.
> > - Q6: Although the authors provided some interesting new results, the experiment still feel a bit thin to me. Regarding models, I feel that it would be important to have results from different categories of models i.e. one GP model (e.g. DUE [1]) and one Posterior Network model (e.g. NatPN [2]) beyond ensembles. It would be pretty convincing to provide these in the tables in the main text. I could not find the details on the ensemble setting for the experiments incl. the number of members. Regarding datasets, the paper currently provides results for datasets where discrete bins are reasonable choices (e.g. ages). The authors mentioned that discrete bins are a natural settings for the used datasets. Since the paper title focuses on regression, it would be important to provide multiple results with datasets with scalar labels (e.g. UCI datasets, depth estimation datasets). As mentioned i the original review, I would also still be interested to have error bars and further uncertainty evaluation like OOD experiments. Action suggestion: I would recommend to add NatPN results (and if possible GP results) and UCI datasets. I would also recommend to add error bars. If possible, it would also be great to have some OOD detection results.
> > - Q1: I feel that the model related works is more complete. Nonetheless, I still feel that some important points that should be addressed. “…and therefore do not work well in the imbalanced setting (Amini et al., 2020; Mi et al., 2022; Charpentier et al., 2022).” The current paper does not provide evidence that the two latter works do not work well in imbalanced settings making this statement questionable. In particular, Mi et al. focus on imbalanced visual regression which looks to be another appropriate baseline. “Meanwhile, Posterior Networks methods Charpentier et al. (2020; 2022); Stadler et al. (2021) consider conjugate distribution, pseudo-count interpretations, posterior updates, and variational losses for fast and high-quality uncertainty estimation. Closest to our work is Deep Evidential Regression (DER) (Amini et al., 2020),”. It is unclear to me why DER is closer to this work than NatPN. As the authors mention, NatPN  consider conjugate distribution, pseudo-count interpretations, posterior updates, and variational losses similarly to VIR. DER. It might be important to better discuss the comparison with NatPN. In particular the “Intuition of Pseudo-Counts for VIR” paragraph and the posterior udpate to be very similar to NatPN. Action suggestion: Revise/explain the mentioned statements and improve the comparison with NatPN.
> > - Q2: The phrasing of the contribution sounds more accurate to me. I feel that the “In order to achieve both desiderata in probabilistic deep imbalanced regression (i.e., performance improvement and uncertainty estimation)” is still confusing. Should it rather be “performance improvement in imbalanced settings” ? Further, I do not see the new discussion on the existing desiderata in the related work section. The new content looks to discuss other model related works right now.
> > - Q3: The additional results are interesting. Based on these experiments, it looks like larger lambda value achieve better results. In practice, why would we not increase the lambda value even more ? Action suggestion: provide results for larger lambda values till results drop.
> >
> > [1] On feature collapse and deep kernel learning for single forward pass uncertainty.
> >
> > [2] Natural Posterior Network: deep bayesian uncertainty for exponential family distributions, ICLR 2022

---

> ### Author Response · Authors · 2022-11-16
> **Thank you and response to your comments [2/2]**
>
> **Q4: "Clarity: The paper is sometimes hard to read. E.g. the paper mentions multiple times “see below” which leaves unclear where to exactly find the missing information. Action suggestion: make explicit reference when pointing to further details."**
>
> This is a good suggestion. We have replaced "see below" with explicit references throughout the paper as suggested.
>
> **Q5: "Bins: It is unclear what is the sensitivity of the method to the number of bins. It was also unclear to me when readin sec. 3.1. what data are used to build the mean representation from the bins. Action suggestion: I would recommend to clarify the bins construction since it is an essential part of the method. I would also be interested in an experiment on the number of bins."**
>
> For fair comparison, we directly follow DIR when constructing bins. For example, in the AgeDB dataset, since the range of the labels (people's ages) is between 1 and 100, it is natural to use 100 bins, each representing one year of age. We have also revised Sec. 3.1 accordingly as suggested.
>
> **Q6: "Experiences: The experiments do not look very extensive to me. They consider only two datasets and a single backbone network. They do not compare to DropOut, Ensemble, GP, or NatPN which would be appropriate baselines. They do not look at common uncertainty estimation metrics like OOD detection scores. They also do not provide error bars which are key to assess the significance of the results. Action suggestion: I would recommend to add at least one dataset (e.g. depth estimation) with another backbone architecture. I would also recommend to add at least Ensemble which is a common and powerful baseline and NatPN which shares many similarities with VIR. I would also recommend to add error bars."**
>
> Thanks for your constructive suggestion. Following your suggestion, we added results for the STS-B dataset, as a third dataset, with the LSTM backbone architecture, as a different backbone, into the revised Appendix. Furthermore, we also added Ensemble to Table 1 & 2 for comparison. For NatPN, we are still tuning the hyperparameters to produce reasonable results, and we will update the response if we can make it before the discussion period ends.
>
>
> Table B.1: New Results on a Third Dataset, STS-B-DIR in Terms of MAE
>
> | Comparison | overall | many | median | few |
> | :-------: | :-----: | :-------: | :---------: | :-------: |
> |Inv| 0.825 | 0.783 | 1.004 | 0.850 |
> |DIR+DER| 0.812 | 0.757 | 1.046 | 0.842 |
> |VIR (Ours)| 0.760 | 0.718 | 0.960 | 0.732 |
> |||||
>
> Table B.2: New Results on a Third Dataset, STS-B-DIR in Terms of Spearman's Correlation
>
> | Comparison | overall | many | median | few |
> | :-------: | :-----: | :-------: | :---------: | :-------: |
> |Inv| 0.723 | 0.678 | 0.530 | 0.685 |
> |DIR+DER| 0.730 | 0.680 | 0.526 | 0.699 |
> |VIR (Ours)| 0.770 | 0.713 | 0.534 | 0.770 |
> |||||
>
>
> Table C.1: New Results on Deep Ensemble in terms of MSE (Lower is Better), Corresponding to Table 1 in the Main Paper
>
> | Comparison | overall | many | median | few |
> | :-------: | :-----: | :-------: | :---------: | :-------: |
> |Deep Ensemble| 100.94 | 79.30 | 129.95 | 249.18 |
> |Vanilla| 101.28 | 78.40 | 131.17 | 256.32 |
> |VIR (Ours)| 86.89 | 77.69 | 96.55 | 145.76 |
> |||||
>
>
> Table C.2: New Results on Deep Ensemble in terms of AUSE (Lower is Better), Corresponding to Table 3 in the Main Paper
>
> | Comparison | overall | many | median | few |
> | :-------: | :-----: | :-------: | :---------: | :-------: |
> |Deep Ensemble| 0.541 | 0.626 | 0.466 | 0.483 |
> |DER| 0.590 | 0.449 | 0.468 | 0.500 |
> |VIR (Ours)| 0.437 | 0.474 | 0.319 | 0.413 |
> |||||
>
> [1] Amini et al., Deep Evidential Regression. NeurIPS 2020

---

### Comment · Area_Chair_VWez · 2022-11-15
**Please engage before the author-reviewer discussion closes**

Dear authors and reviewers,

The first phase of the discussion period is about to close on November 18.

For authors, please make sure to submit your rebuttal by the deadline. Leave some time for the reviewers to read it and respond while you are still allowed to further engage with them. Interactions between authors and reviewers are very important for the quality of the review process, so please make sure to engage.

For reviewers, please try to acknowledge and respond to the authors' rebuttal while the discussion period is still open for them to further interact with you.

Thank you for your participation in the review process!

Best,
The AC

---

### Author Response · Authors · 2022-11-16
**We thank all reviewers for their valuable comments**

We thank all reviewers for their valuable comments. We are glad that they found the problems we identified important/novel/valuable (xTgV, EUDh, ddev, 5JWG), our method/idea interesting/novel/well-motivated/non-trivial (xTgV, EUDh, bYud, ddev, 5JWG), our experiments comprehensive/convincing (xTgV, bYud, ddev), and our writing clear (EUDh). Below we address the reviewers' questions. We have also updated the main paper (with the changed part marked in blue) and the Appendix.

---

### Decision · Program_Chairs · 2023-01-20

**Decision:**

Reject

**Justification For Why Not Higher Score:**

Weaknesses in the experimental validation.

**Justification For Why Not Lower Score:**

N/A

**Metareview: Summary, Strengths And Weaknesses:**

The paper has received mixed reviews, with three reviewers tending to accept (6-6-6) and two reviewers proposing to reject (3-5). The reviewers agree that the paper's contribution is valuable and the problem it addresses is important. Most reviewers have found the paper to be well-written, with clarifications added during the discussion. The most important concern expressed by the reviewers is the experimental validation of the proposed VIR method. Specifically, VIR was assessed on only two age estimation datasets with similar distributions. Preliminary results were obtained on STS-B-DIR during the rebuttal, but the validation would be more convincing and fair against baselines if it were performed on a larger number of datasets, including e.g NYUD2-DIR and SHHS-DIR (as in Yang et al, 2021). Additionally, the reviewers have requested that the authors provide empirical results against missing related work and include error bars in their results. Despite the strong engagement of the authors and the many improvements made during the discussion, I believe that these concerns remain and make the paper not entirely ready for publication.